# Field comparison of dual- and single-spot Aethalometers: equivalent black carbon, light absorption, Ångström exponent and secondary brown carbon estimations

**Liangbin Wu**[1,2,3], **Cheng Wu**[1,2,3], **Tao Deng**[4], **Dui Wu**[1,2,3], **Mei Li**[1,2,3], **Yong Jie Li**[5,6], **and Zhen Zhou**[1,2,3]

[1]Institute of Mass Spectrometry and Atmospheric Environment, Guangdong Provincial Engineering Research Center for Online Source Apportionment System of Air Pollution, Jinan University, Guangzhou 510632, China

[2]Guangdong–Hong Kong–Macau Joint Laboratory of Collaborative Innovation for Environmental Quality, Guangzhou 510632, China

[3]Guangdong MS Institute of Scientific Instrument Innovation, Guangzhou 510530, China

[4]Institute of Tropical and Marine Meteorology, CMA, Guangzhou 510080, China

[5]Department of Civil and Environmental Engineering, Faculty of Science and Technology, University of Macau, Taipa, Macau SAR, China

[6]Department of Ocean Science and Technology, Faculty of Science and Technology, University of Macau, Taipa, Macau SAR, China

**Correspondence:** Cheng Wu (wucheng.vip@foxmail.com)

**Abstract.** The Aethalometer is a widely used instrument for black carbon (BC) mass concentration and light absorption coefficient ($b_{abs}$) measurements around the world. However, field intercomparison of the two popular models, dual-spot (AE33) and single-spot (AE31) Aethalometers, remains limited; in addition, the difference in secondary brown carbon (BrC$_{sec}$) light absorption estimation between the two models is largely unknown. We performed full-year collocated AE33 and AE31 measurements in a megacity in southern China – Guangzhou. The $b_{abs}$ values agree well between the two Aethalometers ($R^2 > 0.95$), with AE33 / AE31 slopes ranging from 0.87 to 1.04 for seven wavelengths. AE33 consistently exhibits lower limits of detection (LODs) than AE31 for time resolutions of 2 to 60 min. The AE33 / AE31 slope for equivalent BC (eBC) was 1.2, implying the need for site-specific post-correction. The absorption Ångström exponent (AAE) obtained from different approaches does not agree very well between the two models, with the biggest discrepancy found in AAE$_{880/950}$. The estimated BrC$_{sec}$ light absorption at 370 nm ($b_{abs370\_BrCsec}$) was calculated using the minimum-$R$-squared (MRS) method for both Aethalometers. The $b_{abs370\_BrCsec}$ comparison yields a slope of 0.78 and an $R^2$ of 0.72 between the two models, implying a non-negligible inter-instrument difference. This study highlights the high consistency in $b_{abs}$ but less so in AAE between AE31 and AE33 and reveals site-specific correction for eBC estimation and non-negligible difference in BrC$_{sec}$ estimation. The results are valuable for data continuity in long-term Aethalometer measurements when transitioning from the older (AE31) to the newer (AE33) model, as anticipated in permanent global-climate and air-quality stations.

## 1 Introduction

Black carbon (BC) and organic carbon (OC) are important carbonaceous aerosol components in the atmosphere, and they play an important role in both global climate and air quality (Bond and Bergstrom, 2006; Li et al., 2021). BC is an important short-lived climate forcer owing to its strong absorption of solar radiation over a wide wavelength range (Bond et al., 2013). A specific group of OC exhibits strong light absorption in the ultraviolet band, and the light absorption decreases with increasing wavelength (Jacobson, 1999), which is later termed brown carbon (BrC) due to its brown-

color appearance (Andreae and Gelencsér, 2006). Previous studies have shown that BrC accounts for about 20 % of the total carbonaceous aerosol light absorption (Feng et al., 2013; Wang et al., 2014; Jo et al., 2016). Light absorption is an important characteristic of BrC, but our current knowledge of BrC is still limited due to the complexity of its chemical composition (Huang et al., 2018). Unlike BC, which was solely emitted from primary sources, BrC can be formed secondarily in the atmosphere (Moise et al., 2015; Laskin et al., 2015). Secondary BrC (BrC$_{sec}$) can be formed via various pathways, e.g., nitration of aromatic compounds (NACs) (Lin et al., 2017), aqueous reactions (Lian et al., 2021; Tang et al., 2016), droplet evaporation (Lee et al., 2013) and iron-catalyzed reactions (Al-Abadleh, 2021). During the atmospheric aging process, the light absorption of BrC can be either enhanced or reduced (Li et al., 2023), depending on whether the chromophores are destroyed (e.g., fragmentation) or re-built (e.g., dimerization) (Jiang et al., 2022).

The filter-based technique (Rosen et al., 1980) has been widely used for aerosol light absorption measurement since its introduction in the early 1980s due to its low operational cost and ease of maintenance (Moosmüller et al., 2009). The Aethalometer (Hansen et al., 1984) is the most commonly used instrument for aerosol light absorption measurement (Lack et al., 2014). Measurement artifacts using the filter-based approach due to the loading effect and multi-scattering effect, however, can bias the results of the Aethalometer (Coen et al., 2010). To tackle these issues, various correction algorithms have been introduced (Weingartner et al., 2003; Arnott et al., 2005; Schmid et al., 2006; Virkkula et al., 2007; Li et al., 2020; Y. Zhang et al., 2021). It is widely recognized that careful post-correction is essential for the accurate light absorption determination by the Aethalometer. Intercomparisons between the Aethalometer and the reference method (e.g., photoacoustic spectroscopy, PAS) have shown that a collocated study is needed to determine the site-specific multi-scattering correction factor ($C_{ref}$) (Arnott et al., 2005; Kim et al., 2018; Zhao et al., 2020). AE31 and AE33 (Aerosol Magee Scientific, CA, USA) are the two most widely used Aethalometers nowadays. Both instruments can provide filter-based measurements at seven wavelengths, but AE33 has the embedded dual-spot technique to perform real-time loading-effect correction (Drinovec et al., 2015), while AE31 requires manual post-correction by the user. Intercomparison of these two models has been carried out (Titos et al., 2015; Rajesh and Ramachandran, 2018; Ferrero et al., 2021). These studies have mainly focused on the equivalent BC (eBC) comparison, while $b_{abs}$ comparison has rarely been reported (Asmi et al., 2021). In addition, two questions remain unanswered. (1) How does the AE33 / AE31 comparison slope vary throughout a long-term measurement period, e.g., a year? Existing AE33–AE31 intercomparisons only cover a few months (Table 1), leaving the seasonality of the intercomparison not well characterized. (2) The inter-instrument bias of BrC$_{sec}$ light absorption ($b_{abs\_BrCsec}$) between AE31

and AE33 has not been investigated. A newly established minimum-$R$-squared (MRS) method for $b_{abs\_BrCsec}$ determination using Aethalometer data (Wang et al., 2019a) has gained popularity in recent Aethalometer studies (Zhu et al., 2021a; Guo et al., 2022; Ivančič et al., 2022; Lei et al., 2023). However, the difference in $b_{abs\_BrCsec}$ determination between AE31 and AE33 remains unknown.

This study aims to fill the aforementioned knowledge gaps. Collocated intercomparison of the AE31 and AE33 was conducted at an urban site in a megacity in southern China (Guangzhou, the capital of Guangdong Province) for 1 full year. Several metrics were characterized and compared between AE31 and AE33, including the limits of detection (LODs), light absorption coefficient, equivalent BC (eBC) mass concentration and absorption Ångström exponent (AAE). The MRS method was used to evaluate the difference in $b_{abs\_BrCsec}$ between AE31 and AE33. The impacts of the data correction schemes, seasonality and temporal resolution were investigated.

## 2  Methods

### 2.1  Observation site and measurement period

The sampling measurements were performed in the Jinan University atmospheric supersite (JNU; 23.13° N, 113.35° E; 40 m above sea level), which lies in the central business district of Guangzhou. The measurement site is on top of a library building and surrounded by a teaching building and residential areas (Liang et al., 2021). The campus is surrounded by the three busiest roads in the city, and traffic emissions are a major source of local air pollutants. Guangzhou is situated in the south of China and is the geographical and business center of Guangdong Province as well. Therefore, the JNU site can represent a typical urban environment in the Pearl River Delta (PRD) region. The measurement period of collocated intercomparison covers 1 year, from April 2021 to March 2022.

### 2.2  Instruments and data correction

Two Aethalometers were compared in this study, a single-spot Aethalometer (AE31, Magee Scientific, CA, USA) and a dual-spot Aethalometer (AE33, Magee Scientific, CA, USA). Both AE31 and AE33 were connected to one PM$_{2.5}$ inlet (Fig. S1 in the Supplement). An inline Nafion dryer (MD-700, Perma Pure, NJ, USA) was used to minimize the impact of relative humidity. Due to the drying capacity of the Nafion dryer, the flow rate of the two Aethalometers was set to be lower than the default value (5 L min$^{-1}$). A lower flow rate can increase the LODs, and that could be an issue for the background sites (e.g., polar regions). Since the eBC concentration in the urban environment is much higher than the LODs of the Aethalometer, the impact is expected to be negligible. The single-spot AE31 was operated at a flow rate

**Table 1.** Summary of existing AE31–AE33 intercomparison studies. W and V in the column "Loading correction" refer to Weingartner and Virkkula correction algorithms, respectively. TS1

| Measurement site | Model | Time base (min) | Flow rate (L min$^{-1}$) | Filter | Loading correction | Period (duration) | Slope (AE33 vs. AE31) | Reference |
|---|---|---|---|---|---|---|---|---|
| Ahmedabad, India (urban) | AE31 | 5 | 3 | Quartz fiber filter $C_{AE31} = 2.14$ | W | Jul 2014–Dec 2014 6 months | eBC$_{880}$ 5 min: 1.06 1 h: 1.02 | Rajesh and Ramachandran (2018) |
| | AE33 | 1 | 3 | Teflon-coated glass fiber $C_{AE33} = 1.57$ | Dual-spot | | | |
| Milan, Italy (urban) | AE31 | 5 | | Quartz fiber filter (Pallflex Q250F) | W | 18 Jan–15 Feb (2018) 1 month | eBC$_{880}$ 5 min: 1.05 | Ferrero et al. (2021) |
| | AE33 | 1 | | Teflon-coated glass fiber (Pallflex T60A20) $C_{AE33} = 1.57$ | Dual-spot | | | |
| Granada, Spain (urban) | AE31 | 5 | | | – | Jun 2014–Jul 2014 2 months | eBC$_{880}$ 5 min: 1.11 | Titos et al. (2015) |
| | AE33 | 1 | | | Dual-spot | | | |
| Pallas, Finland (background) | AE31 | 5 | 4.5 | $C_{AE31} = 3.5$ | V | 19 Jun–17 Jul (2019) 1 month | $b_{abs660}$ 1 h: 0.47 | Asmi et al. (2021) |
| | AE33 | 1 | 5.8 | $C_{AE33} = 1.39$ $C' = 2.52$ | Dual-spot | | | |
| Guangzhou, China (urban) | AE31 | 5 | 2.4 | Quartz fiber filter (Pallflex Q250F) $C_{AE31} = 3.48$ | W and V | Apr 2021–Mar 2022 12 months | eBC$_{880}$ 5 min: 1.18 1 h: 1.20 | This study |
| | AE33 | 1 | 3 | M8060 $C_{AE33} = 1.39$ $C' = 2.1$ | Dual-spot | | $b_{abs880}$ 5 min: 0.85–0.86 1 h: 0.87 | |

of 2.4 L min$^{-1}$ using quartz fiber filter tape (Pallflex, type Q250F). The dual-spot AE33 measurement was conducted at a flow rate of 3 L min$^{-1}$ using filter tape 8060. To evaluate the effectiveness of the Nafion dryer, the ambient RH and RH after the Nafion dryer were compared; see Fig. S2 in the Supplement. The annual average RH was reduced from $60.50 \pm 13.24$ to $45.59 \pm 1.12$ %. More importantly, before drying, half of the data points had an RH higher than 60 %, but after drying, 95 % of the data points had an RH lower than 60 % (Fig. S2a). Additionally, the diurnal fluctuations were effectively minimized after Nafion drying (Fig. S2b). These results suggested that the RH of the sample air was well controlled before entering the two Aethalometers. The data acquisition time base was 5 and 1 min for AE31 and AE33, respectively. Both AE31 and AE33 were set to advance the filter tape to a new spot when the light attenuation (ATN) at 370 nm reached 100. Routine maintenance procedures suggested by Cuesta-Mosquera et al. (2021) were implemented in this study. The optical chamber of the two Aethalometers was carefully cleaned before the collocated experiment, and this was repeated every 3 months. Flow verification and calibrations of the two Aethalometers were conducted every 3 months using an external flow meter (Bios Defender 520H,

Mesa Labs, CO, USA). Blank tests and leak tests were performed monthly for the two Aethalometers.

### 2.2.1 Single-spot Aethalometer – AE31

The AE31 measures light attenuation (ATN) at seven wavelengths (370, 470, 520, 590, 660, 880 and 950 nm) through a particle-laden filter. ATN can be calculated by the Beer–Lambert law:

$$ATN = -100 \cdot \ln(I/I_0), \tag{1}$$

where $I$ and $I_0$ are the intensities of light transmitted through the particle-laden filter and particle-free filter, respectively. The aerosol sample is continuously deposited on the filter, leading to the increase in ATN over time. The light attenuation coefficient ($b_{ATN}$) for particles collected on the filter tape is defined as follows:

$$b_{ATN} = \frac{A}{F} \cdot \frac{\Delta ATN}{\Delta t}, \tag{2}$$

where $A$ is the sample spot area, $F$ is the aerosol flow rate and $\Delta ATN$ is the change in ATN over a time period $\Delta t$. It is worth noting that $b_{ATN}$ differs from the aerosol light absorption coefficient ($b_{abs}$) because it is determined by the ATN through the particle-laden filter, and the discrepancy can be

reconciled by different algorithms (Coen et al., 2010). Then $eBC_{raw}$ can be calculated from

$$eBC_{raw} = \frac{b_{ATN}}{\sigma_{ATN}}. \tag{3}$$

Here $\sigma_{ATN}$ is the conversion factor between $b_{ATN}$ and $eBC$, which is obtained from the regression slope between $b_{ATN}$ and elemental carbon (EC) by the EGA (evolve gas analysis) method (Gundel et al., 1984). Developed by Lawrence Berkeley National Laboratory (LBNL), the EGA method (Ellis et al., 1984) was commonly used from the 1980s to 1990s (Ip et al., 1984; Turner and Hering, 1990; Young et al., 1994) and has been less popular in recent years. Since the BC of Aethalometer was calibrated to the LBNL–EGA EC, differences in EC analysis protocols lead to a disagreement between eBC and other popular EC methods (e.g., NIOSH and IMPROVE) used nowadays. In general, it is recognized that eBC is usually higher than that according to NIOSH (Jeong et al., 2004) but lower than that according to IMPROVE (Watson and Chow, 2002). The values of $\sigma_{ATN}$ at seven wavelengths recommended by the manufacturer can be found in Table S1 in the Supplement.

In this study, two data correction algorithms were applied and compared, namely Weingartner (Weingartner et al., 2003) and Virkkula (Virkkula et al., 2007). Both Weingartner and Virkkula corrections were implemented for AE31 data using an Igor Pro-based toolkit ("Aethalometer data processor") (Wu et al., 2018).

The Weingartner scheme is defined as follows:

$$eBC_{corrected} = \frac{eBC_{raw}}{R(ATN)}, \tag{4}$$

$$R(ATN) = \left(\frac{1}{f} - 1\right) \cdot \frac{\ln(ATN) - \ln(10)}{\ln(50) - \ln(10)} + 1, \tag{5}$$

where $eBC_{corrected}$ is mass concentration after loading correction. $R(ATN)$ is the correction function for the loading effect, and $f$ is the empirical filter loading-effect compensation parameter.

The Virkkula correction can be calculated as follows:

$$eBC_{corrected} = (1 + k \cdot ATN) \cdot eBC_{raw}, \tag{6}$$

where $k_i$ is the loading-effect compensation parameter for the $i$th sampling spot that can be calculated from the following equation:

$$k_i = \frac{eBC_{raw}(t_{i+1,first}) - eBC_{raw}(t_{i,last})}{ATN(t_{i,last}) \cdot eBC_{raw}(t_{i,last})} , \tag{7}$$
$$-ATN(t_{i+1,first}) \cdot eBC_{raw}(t_{i+1,first})$$

where $eBC_{raw}$ is the raw BC concentration before correction, ATN is light attenuation, $t_{i,last}$ refers to the last measurement data for filter spot $i$ and $t_{i+1,first}$ refers to the first measurement data for the next filter spot. An example of AE31 data

before and after correction is shown in Fig. S3 in the Supplement. It is very clear that data discontinuity during filter advance was effectively minimized by both algorithms.

Once $eBC_{corrected}$ was obtained by either the Weingartner or the Virkkula algorithm, the corresponding light absorption coefficient ($b_{abs}$) can be back-calculated from

$$b_{abs}(AE31) = \frac{eBC_{corrected} \cdot \sigma_{ATN}}{C_{AE31}}, \tag{8}$$

where $b_{abs}$ is the aerosol light absorption coefficient in the air and $C_{AE31}$ is the multiple-scattering parameter, whose value depends on the filter material and mixing state of the particles (coating thickness). For example, $C_{AE31} = 3.6 \pm 0.6$ was observed in the organic coating experiment using a quartz filter (Weingartner et al., 2003). In this study, $C_{AE31} = 3.48$ was adopted for AE31 according to a previous intercomparison study between using an Aethalometer and PAS in Guangzhou (Wu et al., 2013). This value is similar to those recommended by the guidelines from the Global Atmosphere Watch Programme ($C_{AE31} = 3.5$) (World Meteorological Organization, 2016).

### 2.2.2 Dual-spot Aethalometer – AE33

AE33 collects particles on two spots with different flow rates, leading to a different ATN increase over time (Drinovec et al., 2015); thus the $b_{ATN}$ of the two spots can be calculated from

$$b_{ATN1} = \frac{A}{F1} \cdot \frac{\Delta ATN1}{\Delta t}, \tag{9}$$

$$b_{ATN2} = \frac{A}{F2} \cdot \frac{\Delta ATN2}{\Delta t}, \tag{10}$$

where $A$ is the sample spot area and $F_1$ and $F_2$ are the flow rates at the two sampling spots, with a ratio of $2:1$. $\Delta ATN1$ and $\Delta ATN2$ are the changes in ATN over a time period $\Delta t$ at the two spots. The raw BC concentration can be calculated from

$$eBC1_{raw} = \frac{b_{ATN1}}{C_0 \cdot \sigma_{air}}, \tag{11}$$

$$eBC2_{raw} = \frac{b_{ATN2}}{C_0 \cdot \sigma_{air}}. \tag{12}$$

$C_0$ is the multiple-scattering parameter provided by the manufacturer, which strongly depends on the material of the filter tape. For example, $C_0 = 2.14$ should be applied for a quartz filter and $C_0 = 1.57$ was used for tape model 8020 and 8050 (Drinovec et al., 2015), respectively. In contrast, $C_0 = 1.39$ should be applied for tape 8060 (Magee Scientific, 2017), which is the case for the current study. Here $\sigma_{air}$ is the conversion factor between $b_{abs}$ and eBC, and the values of $\sigma_{air}$ of AE33 at seven wavelengths recommended by the manufacturer can be found in Table S1. It should be noted that the physical meaning of $\sigma_{air}$ in Eqs. (11) and (12) is different

from $\sigma_{ATN}$ in Eq. (3). In AE31, $\sigma_{ATN}$ converts light absorption on the filter ($b_{ATN}$) to eBC, while in AE33, $\sigma_{air}$ converts light absorption in the air ($b_{abs}$) to eBC. The relationship between $\sigma_{air}$ and $\sigma_{ATN}$ can be written as $\sigma_{ATN} = C_0 \cdot \sigma_{air}$. The $\sigma_{air}$ value was derived from the historical $\sigma_{ATN}$ value using $C_0 = 2.14$, as shown in Table S1 (Drinovec et al., 2015).

With known ATN and raw eBC concentrations of the two spots, the corrected eBC can be calculated from the dual-spot equations (Drinovec et al., 2015):

$$eBC1_{raw} = eBC_{corrected} \cdot (1 - k \cdot ATN1), \quad (13)$$

$$eBC2_{raw} = eBC_{corrected} \cdot (1 - k \cdot ATN2). \quad (14)$$

Here $k$ is the loading-effect compensation parameter. It should be noted that $k$ in the Virkkula correction of AE31 data is a constant for all data points within each tape advance cycle, while $k$ in the dual-spot correction of the AE33 data is a variable that can be calculated for every data point. By solving the two equations, both $k$ and eBC$_{corrected}$ can be determined. An example of AE33 data before and after correction is shown in Fig. S3. It is very clear that data discontinuity during filter advance was successfully minimized by the dual-spot algorithms.

Once eBC$_{corrected}$ is determined, $b_{abs}$ can be back-calculated from

$$b_{abs}(AE33) = \frac{eBC_{corrected} \cdot \sigma_{air}}{H}. \quad (15)$$

Here $\sigma_{air}$ is the conversion factor between $b_{abs}$ and eBC; the values of $\sigma_{air}$ of AE33 at seven wavelengths recommended by the manufacturer can be found in Table S1. As suggested by a previous study in Guangzhou, the $C_0$ recommended by the manufacturer is not sufficient to achieve a 1 : 1 slope with the reference instrument; thus a second correction factor (also known as a harmonization factor) $H = 2.1$ was introduced (Qin et al., 2018). A study in central Oregon, USA, also found that $C_0 = 1.57$ by default is too low, and $C_{AE33} = 4.35$ was recommended (Laing et al., 2020). Therefore, a final correction factor of $C_{AE33} = C_0 \cdot H = 2.919$ (filter tape 8060) is used for AE33 in this study. This value is very close to the value ($C_{AE33} = 2.9 \pm 0.4$, filter tape 8060) found in eastern China (Zhao et al., 2020) (Table S2 in the Supplement) but slightly higher than those values used by European ACTRIS measurement network ($C_{AE33} = C_0 \cdot H = 1.39 \cdot 1.76 = 2.45$, filter tape 8060) (Savadkoohi et al., 2023).

### 2.2.3 Absorption Ångström exponent

Absorption Ångström exponent (AAE) is a useful parameter that characterizes the wavelength dependence of particle absorption. AAE can be calculated from two approaches, and previous studies have shown that the two approaches can lead to different results (Lack and Cappa, 2010; Helin et al., 2021). The first approach involves calculations using two wavelengths:

$$AAE_{\lambda 1/\_\lambda 2} = -\frac{\ln(b_{abs\_\lambda 1}) - \ln(b_{abs\_\lambda 2})}{\ln(\lambda_1) - \ln(\lambda_2)}, \quad (16)$$

where $AAE_{\lambda 1/\_\lambda 2}$ is the absorption Ångström exponent and $b_{abs\_\lambda 1}$ and $b_{abs\_\lambda 2}$ are the light absorption coefficients at wavelength $\lambda_1$ and wavelength $\lambda_2$, respectively.

The second approach utilizes all available wavelengths in a specific range by power-law curve fitting. Detailed calculation examples are given in Text S1 in the Supplement. To distinguish the AAE values calculated from these two approaches, different notations were used. For example, $AAE_{370/950}$ refers to the AAE calculated by Approach 1, while $AAE_{370-950}$ represents the AAE determined by Approach 2 using all wavelength data between 370 and 950 nm (seven wavelengths for both AE31 and AE33). This notation emphasizes that "/" represents the separator of the two wavelengths in Approach 1 and "–" denotes the range of wavelength in Approach 2.

### 2.2.4 Light absorption of secondary brown carbon

The secondary brown carbon light absorption at 370 nm ($b_{abs370\_BrCsec}$) was calculated using the minimum-$R$-squared (MRS) method (Wu and Yu, 2016; Wang et al., 2019a). First, the $b_{abs}$ can be divided into two parts: secondary brown carbon light absorption ($b_{abs\_\lambda\_BrCsec}$) and non-secondary brown carbon light absorption ($b_{abs\_\lambda\_other}$) at wavelength $\lambda$:

$$b_{abs\_\lambda} = b_{abs\_\lambda\_BrCsec} + b_{abs\_\lambda\_other}, \quad (17)$$

where $b_{abs\_\lambda}$ is the total light absorption at wavelength $\lambda$ from direct measurements. It is generally believed that the light absorption of BrC is negligible at the wavelength of 880 nm, and BrC$_{sec}$ is secondarily generated during the aging process. Therefore it is assumed that BrC$_{sec}$ is not correlated with BC. Based on this assumption, the light absorption at 880 nm can be used as a tracer to characterize $b_{abs\_\lambda\_other}$ at shorter wavelengths (e.g., 370 to 660 nm):

$$b_{abs\_\lambda\_other} = \left(\frac{b_{abs\_\lambda}}{b_{abs\_880}}\right)_{pri} \times b_{abs\_880}, \quad (18)$$

where $b_{abs\_880}$ is the light absorption coefficient at 880 nm. The key parameter here is the primary ratio ($b_{abs\_\lambda}/b_{abs\_880}$)$_{pri}$, which can be calculated using the Igor-based MRS toolkit (Wu and Yu, 2016). As a result, $b_{abs\_\lambda\_BrCsec}$ can be determined as follows:

$$b_{abs\_\lambda\_BrCsec} = b_{abs\_\lambda} - \left(\frac{b_{abs\_\lambda}}{b_{abs\_880}}\right)_{pri} \times b_{abs\_880}. \quad (19)$$

### 2.2.5 Data analysis and visualization

Several data analysis and visualization toolkits developed in our group were used in this study, including "Scatter Plot", "Histbox" and "Aethalometer data processor".

*Scatter Plot.* Conventional ordinary least squares (OLS) assumes that independent variables ($X$) are error-free. However, for inter-instrument comparison studies, $X$ and $Y$ (from two instruments) usually have comparable degrees of uncertainty. In this case, linear regression by OLS should be avoided as it leads to a biased slope and intercept. To account for uncertainties in both $X$ and $Y$, an error-in-variables linear regression technique, weighted orthogonal distance regression (WODR), was applied in this study, implemented by the Igor-based toolkit "Scatter Plot" (Wu and Yu, 2018). A free download of "Scatter Plot" can be found at https://doi.org/10.5281/zenodo.832416 (Wu, 2020a).

*Histbox.* A handy tool enables batch plotting for histogram and boxplots with specific optimization for atmospheric science (e.g., batch plotting by year–season–month, by hour, by day of week, by user-defined strings). The Igor-based "Histbox" toolkit (Wu et al., 2018) also provides data averaging and alignment functions, which are common steps in atmospheric data processing (e.g., integrating data from various instruments with different timescales). Its comprehensive data sorting, grouping and screening features ensure efficient data visualization. A free download of "Histbox" can be found at https://doi.org/10.5281/zenodo.832405 (Wu, 2020b).

*Aethalometer data processor.* Data acquired from filter-based measurements such as legacy Aethalometer models (AE31 and AE20) need careful correction due to their inherent systemic error, i.e., the filter matrix effect, scattering effect and loading effect. This toolkit (Wu et al., 2018) provides a user-friendly interface to implement Weingartner et al. (2003) and Virkkula et al. (2007) algorithms for Aethalometer data correction. QA/QC features are also provided, including statistics of sensor voltage. A free download of "Aethalometer data processor" can be found at https://doi.org/10.5281/zenodo.832403 (Wu, 2020c).

## 3 Results and discussions

### 3.1 eBC concentration comparison

The limits of detection (LODs) of eBC of the AE31 and AE33 Aethalometers were derived from 3 times the standard deviation of the blank measurements. The blanks were obtained by placing a HEPA filter upstream of the two Aethalometers. During the experiment, the time bases of both instruments were set to their highest time resolution (1 s for AE33 and 2 min for AE31) with a sampling flow rate of $5 \, \text{L} \, \text{min}^{-1}$. In total, blank data spanning 96 h were obtained for both Aethalometers.

To investigate the LODs of eBC at different time resolutions, data averaging was performed at various time bases as summarized in Table 2. The LOD decreases with an increased data averaging interval as expected. For example, the 370 nm eBC LOD of AE33 was $533.67 \, \text{ng} \, \text{m}^{-3}$ at 1 s

and can be reduced to $82.17 \, \text{ng} \, \text{m}^{-3}$ if the time base is changed to 1 min. For the same time base, eBC LOD increases with longer wavelengths. As the newer model, AE33 exhibits a lower eBC LOD at all wavelengths. For example, 370 nm eBC LODs were 75.42 and $197.01 \, \text{ng} \, \text{m}^{-3}$ at 2 min for AE33 and AE31, respectively (Fig. 1). The AE33–AE31 LOD difference becomes smaller for longer time intervals. For example, the 370 nm LODs at 60 min were 14.25 and $20.82 \, \text{ng} \, \text{m}^{-3}$ for AE33 and AE31, respectively (Fig. S4a in the Supplement). In addition, the LOD difference between 370 and 950 nm of AE33 (e.g., 75.42 vs. $99.72 \, \text{ng} \, \text{m}^{-3}$ at 2 min) was much smaller than that of AE31 (e.g., 197.01 vs. $789.99 \, \text{ng} \, \text{m}^{-3}$ at 2 min) as shown in Table 2. In other words, the AE33 LOD improvement was more pronounced at longer wavelengths (e.g., 96.57 vs. $730.68 \, \text{ng} \, \text{m}^{-3}$ at 880 nm and 2 min for AE33 and AE31, respectively) as shown in Table 2 and Fig. 1f. In summary, the LOD performance was significantly improved for AE33, especially for the infrared (IR) channels (880 and 950 nm), which are commonly used for reporting eBC concentrations. The detection limits of Aethalometers are wavelength-dependent because the LED of each wavelength may have different characteristics in terms of the light intensity stability, background noise and detector response. The electronic component can also affect the LODs of the Aethalometer. A study using AE51 by Ning et al. (2013) showed that the LOD with a 5 V DC power supply was 5 times the LOD with a battery power supply. The improved LOD of AE33 compared to AE31 is due to a combination of advances in LED stability, flow control, leakage reduction, optical chamber design and electronics (Drinovec et al., 2015).

Among the seven wavelengths, 880 nm is recognized as the standard wavelength for reporting eBC concentration, since the interference of BrC and dust with BC determination can be minimized in the IR range. We therefore discuss the eBC comparison at 880 nm in this section. To maintain consistent long-term eBC measurement results, the older model (AE31) was selected as the reference instrument. For this reason, AE31 data were set as the $X$ variable and the AE33 data were set as the $Y$ variable in the linear regression (Fig. 2). To investigate the effect of data correction schemes, AE31 results from both Virkkula and Weingartner corrections were included in the comparison. As shown in Fig. 2, hourly eBC from the two Aethalometers agree very well, with high $R^2$ values (0.96–0.97) and a slope of 1.2 (Fig. 2a and b). The 5 min data yield similar results, with a slope of 1.18 and $R^2$ of 0.91 (Fig. 2d and e). The annual average eBC by AE31 for 1 h and 5 min data were $1.95 \pm 1.12$ and $1.96 \pm 1.18 \, \mu\text{g} \, \text{m}^{-3}$, respectively. The annual average eBC by AE33 for 1 h and 5 min data were identical ($2.35 \, \mu\text{g} \, \text{m}^{-3}$). These results imply that the inter-instrument slope and annual mean of eBC are not sensitive to the time resolution of the data. The inter-instrument eBC slope obtained in this study is higher than those slopes found in previous studies, as summarized in Table 1. A slope of 1.11 was observed in a 2-month study at an

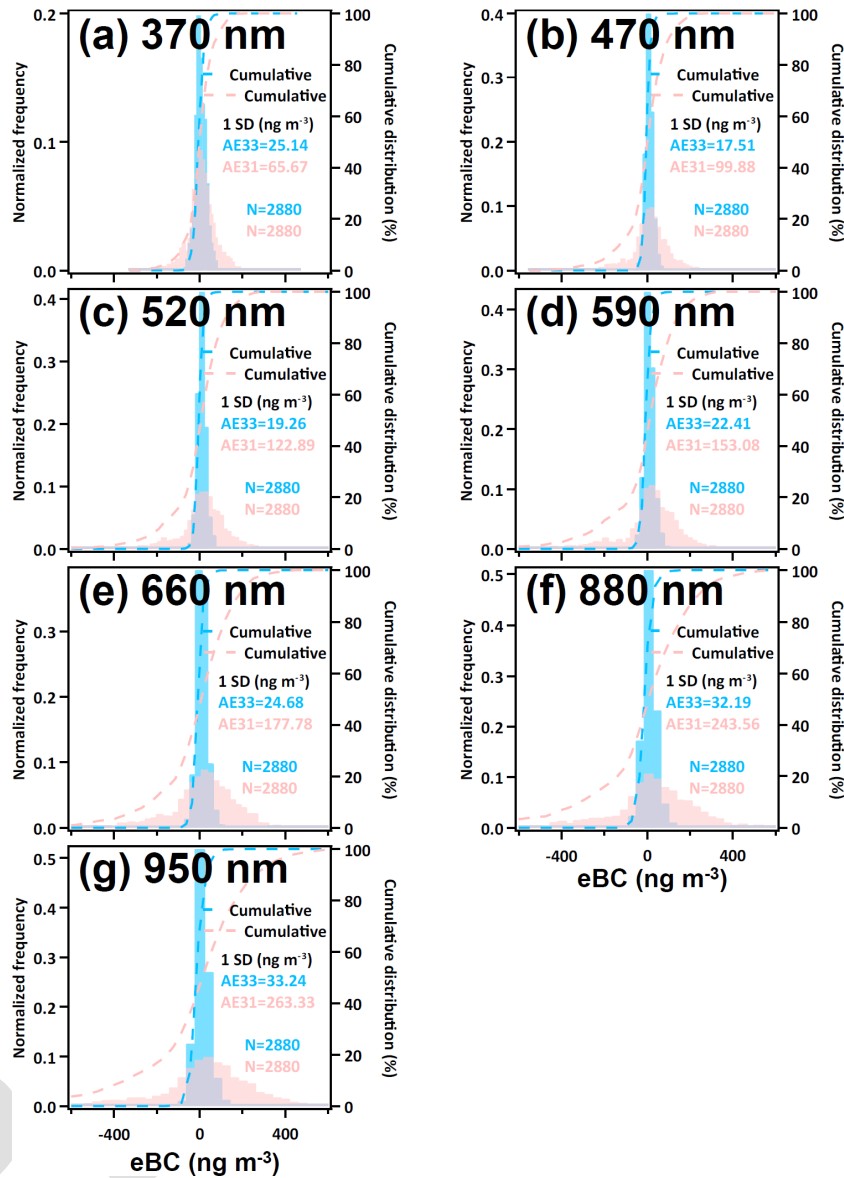

**Figure 1.** The frequency distributions of blank measurements of AE31 and AE33 at the time base of 2 min. The red histograms represent AE31, and AE33 results are shown in blue histograms. Panels **(a)**–**(g)** correspond to 370, 470, 520, 590, 660, 880 and 950 nm, respectively.

urban site in Spain. Similar results were reported elsewhere, e.g., a slope of 1.02 from a 6-month study in urban India (Rajesh and Ramachandran, 2018) and a slope of 1.05 from a 1-month study in urban Italy (Ferrero et al., 2021). The eBC differences between AE33 and AE31 were associated with factors like hardware design and filter type. Along with the site-dependent aerosol type and mixing state, these factors could lead to site-dependent eBC differences between AE33 and AE31. According to the technical notes of the manufacturer (Magee Scientific, 2017), the slope of eBC by 8060 or 8020 filter varied by different locations, e.g., Beijing (0.82), Bengaluru (0.87), Paris (0.93) and Berkeley (0.94). The filter used by AE31 (quartz fiber filter, Pallflex Q250F) is very

different from the filter used by AE33 (8060) in terms of material and optical properties. Likewise, the site-dependent filter difference could contribute to the site-dependent AE33–AE31 difference. Along with the variations in the AE33–AE31 difference reported in previous studies (Table 1), the $\sim 20\%$ bias in the slope found in this study suggests that the AE33 / AE31 slope could be site-dependent. As a result, post-adjustment is needed to obtain consistent results from the two Aethalometers.

Considering the operationally defined nature of eBC and the large number of historical eBC data accumulated by legacy-type Aethalometers, it would be more appropriate to align the eBC from the newer model to the eBC from legacy-

**Table 2.** The eBC LODs ($\mathrm{ng\,m^{-3}}$) of AE31 and AE33 at different time bases at seven wavelengths. $N$ refers to the number of data points.

| Model | Time base | 370 nm | 470 nm | 520 nm | 590 nm | 660 nm | 880 nm | 950 nm | $N$ |
|---|---|---|---|---|---|---|---|---|---|
| AE33 | 1 s | 533.67 | 601.89 | 662.34 | 706.68 | 807.51 | 1177.56 | 1234.95 | 345 600 |
| | 1 min | 82.17 | 62.37 | 68.28 | 79.98 | 86.07 | 112.71 | 118.17 | 5760 |
| | 2 min | 75.42 | 52.53 | 57.78 | 67.23 | 74.04 | 96.57 | 99.72 | 2880 |
| | 4 min | 66.06 | 44.43 | 48.51 | 56.88 | 62.55 | 81.75 | 83.55 | 1440 |
| | 10 min | 37.38 | 24.87 | 27.30 | 31.86 | 34.77 | 45.66 | 46.05 | 576 |
| | 60 min | 14.25 | 9.00 | 9.78 | 11.13 | 11.58 | 14.97 | 14.88 | 96 |
| AE31 | 2 min | 197.01 | 299.64 | 368.67 | 459.24 | 533.34 | 730.68 | 789.99 | 2880 |
| | 4 min | 164.25 | 244.29 | 299.19 | 375.51 | 431.34 | 594.42 | 645.72 | 1440 |
| | 10 min | 77.07 | 115.83 | 135.69 | 165.24 | 195.99 | 263.4 | 290.58 | 576 |
| | 30 min | 31.17 | 45.39 | 48.42 | 55.62 | 64.08 | 79.53 | 105.93 | 192 |
| | 60 min | 20.82 | 28.29 | 30.78 | 36.39 | 43.29 | 52.20 | 71.10 | 96 |

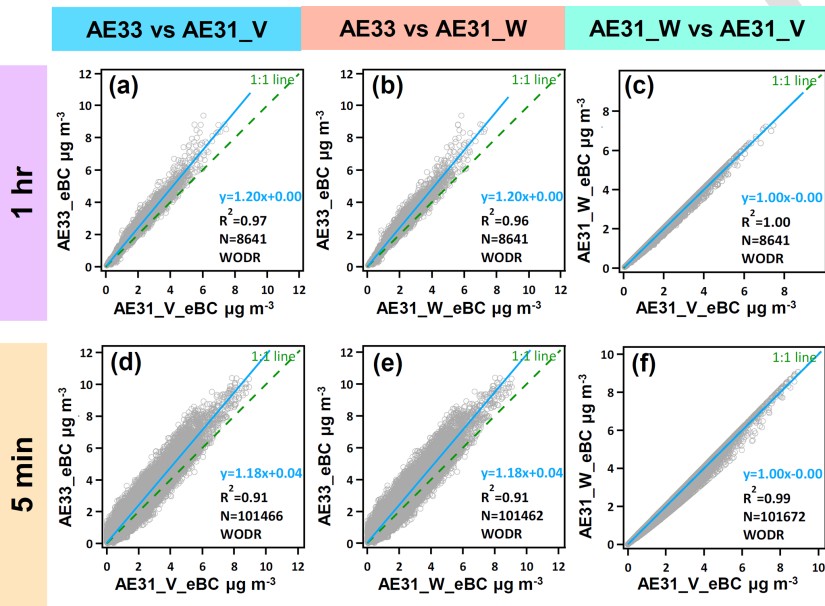

**Figure 2.** Comparisons of 5 min and 1 h eBC mass concentrations at 880 nm between AE33 and AE31. AE31_V_eBC and AE31_W_eBC are the eBC mass concentrations of AE31 corrected by the Virkkula and Weingartner algorithms, respectively. AE33_eBC represents the eBC mass concentration of AE33 at 880 nm.

type Aethalometers to maintain the consistency of the historical data of AE31. For these reasons, the eBC mass concentration of AE33 Aethalometer was further adjusted by a second correction factor ($C_{\mathrm{eBC}}$, 1.20 and 1.18 for 1 h and 5 min data, respectively), which is the slope obtained in Fig. 2:

$$eBC_{2nd\_cor} = \frac{eBC}{C_{\mathrm{eBC}}}. \tag{20}$$

After the eBC correction, the annual averages of the two instruments were also in good agreement as shown in Fig. S5 in the Supplement. The annual mean eBC obtained from 1 h AE33 data were $2.35 \pm 1.37$ and $1.96 \pm 1.14\,\mathrm{\mu g\,m^{-3}}$, respectively, before and after eBC correction (Table S3 in the Supplement). The latter value agrees well with the AE31 annual average ($1.95 \pm 1.12\,\mathrm{\mu g\,m^{-3}}$, Table S3). In summary,

fine tuning of the AE33 data by applying a site-specific inter-instrument correction factor ($C_{\mathrm{eBC}}$) is needed to maintain the consistency of the historical data from AE31.

### 3.2 Intercomparison of light absorption coefficient

The light absorption coefficient by the two Aethalometers was compared, including AE31 data corrected by the Virkkula (AE31_V_$b_{\mathrm{abs}}$) and Weingartner (AE31_W_$b_{\mathrm{abs}}$) algorithms and AE33 results (AE33_$b_{\mathrm{abs}}$). The differences in the $b_{\mathrm{abs}}$ annual mean between AE33 and AE31 are small across the seven wavelengths (Table S4 in the Supplement). The year-long hourly $b_{\mathrm{abs}}$ values of the two Aethalometers agree well, as evidenced by the high $R^2$ (0.95–1) and close-to-unity slope (0.87–1.04) as shown in Fig. 3. The slopes for

AE33_$b_{abs}$ vs. AE31_V_$b_{abs}$ vary slightly by wavelengths, ranging from 0.87 to 0.97 (Fig. 3). The AE33–AE31 agreement on $b_{abs660}$ observed in this study (slope = 0.89, Fig. 3m) is much better than a previous study (slope = 0.47) conducted in the polar region (Asmi et al., 2021). Considering the $b_{abs}$ level ($\sim 0.1\,\mathrm{Mm^{-1}}$) in the study by Asmi et al. (2021) is close to the LODs of Aethalometers and 2 orders of magnitude lower than the $b_{abs}$ level of the current study, it is not surprising that the higher loadings of light-absorbing aerosol particles of this study favor a better AE33–AE31 agreement on $b_{abs}$. The comparisons from the AE33_$b_{abs}$ vs. AE31_W_$b_{abs}$ data yield similar results (slope 0.87–1.04), which implies that inter-instrument comparison results between AE33 and AE31 are not sensitive to the data correction schemes used for AE31. The inter-instrument $b_{abs}$ divergence at different wavelengths may be associated with the difference in filter tape material and optical chamber design of Aethalometers, as well as the optical properties of aerosols. A study by Yus-Díez et al. (2021) in Spain found that the $C_{ref}$ value at IR wavelengths was higher than those values at UV wavelengths when the single-scattering albedo (SSA) was higher than a specific threshold, which was attributed to the presence of dust from the Sahara. But if SSA was lower than a specific threshold, $C_{ref}$ exhibited no dependence on the wavelength. A study in Italy (Bernardoni et al., 2021) found that $C_{ref}$ strongly depended on filter tape material and the wavelength dependence is small. Thus these factors could contribute to the inter-instrument $b_{abs}$ difference at different wavelengths. On-site determination of wavelength-specific $C_{ref}$ is expected to further improve the AE33 vs. AE31 agreement in $b_{abs}$. However, the wavelengths of existing commercially available multi-wavelength reference instruments (e.g., PAX, PAAS and DPAS) do not fully cover the range of Aethalometers (370–950 nm), which makes such a study very challenging. Nonetheless, the $b_{abs}$ agreement between AE31 and AE33 in this study is good enough despite a single $C_{ref}$ being adopted from previous studies (Table S2).

To investigate the effect of the data correction algorithm on AE31 data, a comparison of AE31_W vs. AE31_V was also conducted, as shown in Fig. 3. The close-to-unity slopes (0.97–0.99) were observed from 470 to 950 nm, while the slope at 370 nm exhibits a slight bias (0.93). The high $R^2$ (0.99–1) values found at all seven wavelengths suggest that the results from both algorithms agree very well.

Besides hourly results, $b_{abs}$ intercomparison was also conducted for 5 min data and yields similar slopes (0.85–1.03), as illustrated in Fig. S6 in the Supplement. The $R^2$ values (0.89–1) of 5 min data were slightly lower than those of 1 h data sets, which was as expected, since the increase in data averaging interval can lead to higher inter-instrument $R^2$ values.

To investigate the monthly variability in AE33 vs. AE31 $b_{abs}$ comparison, linear regression was also performed for individual months as shown in Fig. 4. The monthly trend of slope variations between AE33 vs. AE31_V and AE33 vs.

AE31_W is identical, being higher in October and November and lower in July. In addition, no monthly variations were observed for the AE31W vs. AE31_V comparison. These results imply that monthly variations in the AE33 vs. AE31 slopes are not sensitive to the correction schemes used for the AE31 data. The monthly trend of slope variations was similar between 5 min and 1 h data, and the main difference is the increased $R^2$ values in the 1 h data (Tables S5 and S6 in the Supplement). The maximum relative slope deviation of individual months compared to the annual average was 12.62 % and 16.28 % for 370 and 880 nm, respectively (Table S7 in the Supplement). The results suggest that the $b_{abs}$ comparison between AE33 and AE31 exhibits observable monthly variations, but the degree of monthly variations is relatively small.

To explore the $b_{abs}$ uncertainty due to $C$ values, a sensitivity test on different $C$ values was performed. According to a recent comparison study between PAS and Aethalometer (Zhao et al., 2020), the $C$ value deviation was found to be ±0.4 in the North China Plain. Thus, a deviation of ±0.4 with an interval of 0.1 was used for the sensitivity test. The results are shown in Table S8 in the Supplement. With a deviation of ±0.4 for the $C$ values of AE33, the corresponding AE33 / AE31 slopes of $b_{abs}$ range from 0.81 to 1.11, which provides a rough estimation of $b_{abs}$ uncertainty due to $C$ values. It is also worth noting the uneven agreement of $b_{abs}$ between different wavelengths. This issue is related to the use of a single $C$ value across all seven wavelengths, which is due to the absence of multi-wavelength PAS. By far, the wavelength coverage of PAS instruments remains limited, e.g., two wavelengths (405 and 880 nm) (Lewis et al., 2008) and four wavelengths (405 to 660 nm) (Schnaiter et al., 2023). This issue cannot be fully resolved before the emergence of PAS that can fully cover the wavelengths of Aethalometers from 370 to 950 nm.

## 3.3 Inter-instrument comparison of absorption Ångström exponent

The AAE is widely recognized as a useful indicator for differentiating BC and BrC (Zheng et al., 2021; Wang et al., 2021a), as well as mixing state (Lack and Langridge, 2013). However, inter-instrument comparisons of AAE between different models of Aethalometers have rarely been reported. In addition, a previous study has shown that using two approaches (Approach 1, using two wavelengths, and Approach 2, power-law fitting using all wavelengths) for AAE determination may impact the results and even affect interpretation (Lack and Cappa, 2010). To gain further insights into AAE determination, the AAE values obtained by the two approaches (Approach 1: $AAE_{470/660}$, $AAE_{370/880}$, $AAE_{880/950}$, $AAE_{370/950}$; Approach 2: $AAE_{370-950}$) using data from AE31_V, AE31_W and AE33 were compared, as shown in Fig. 5 and Table 3.

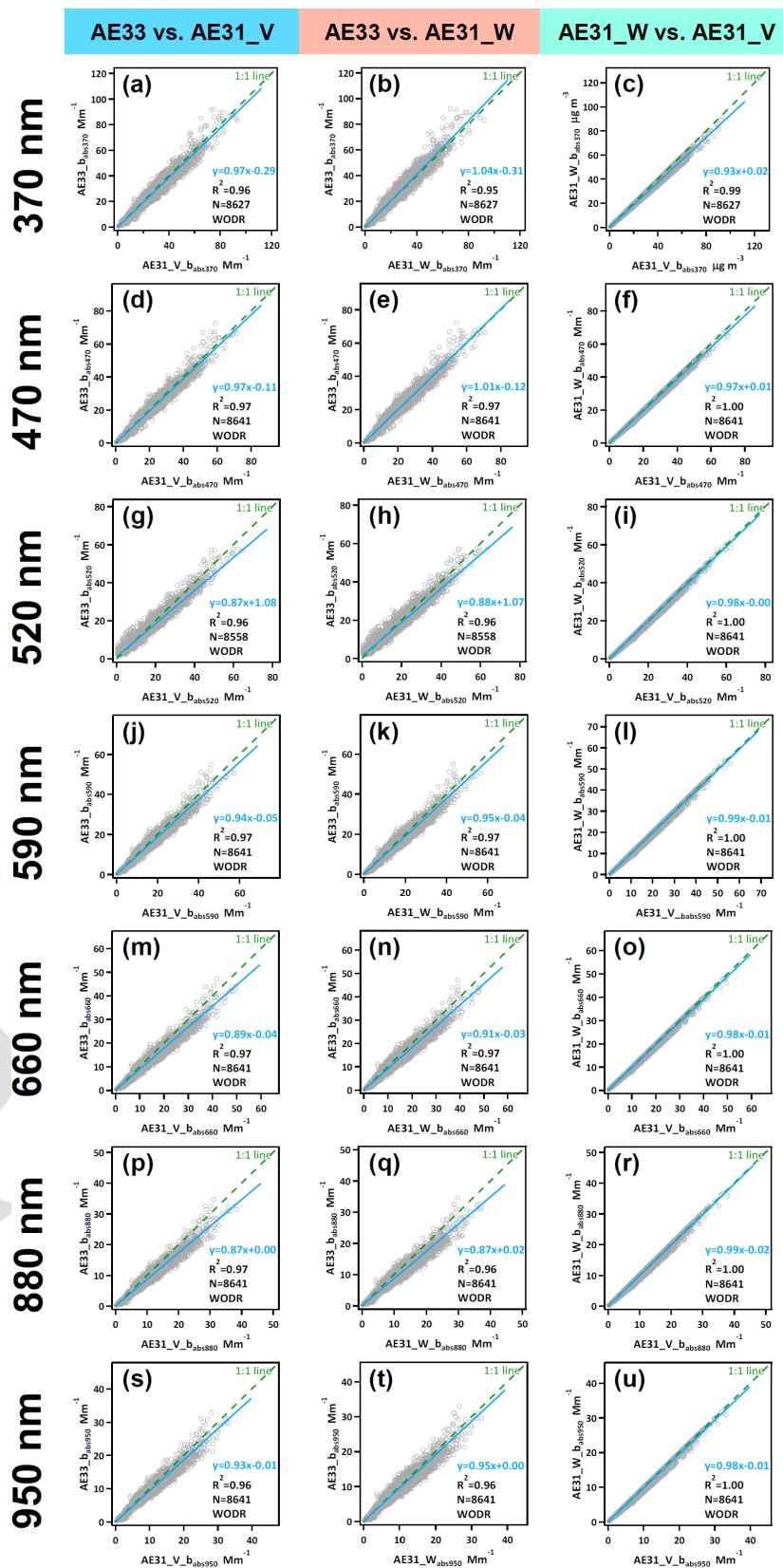

**Figure 3.** Comparison of hourly light absorption coefficient between AE33 and AE31 at 370, 470, 520, 590, 660, 880 and 950 nm. AE31_V_$b_{abs}$ and AE31_W_$b_{abs}$ represent light absorption coefficients of AE31 corrected by Virkkula and Weingartner algorithms, respectively.

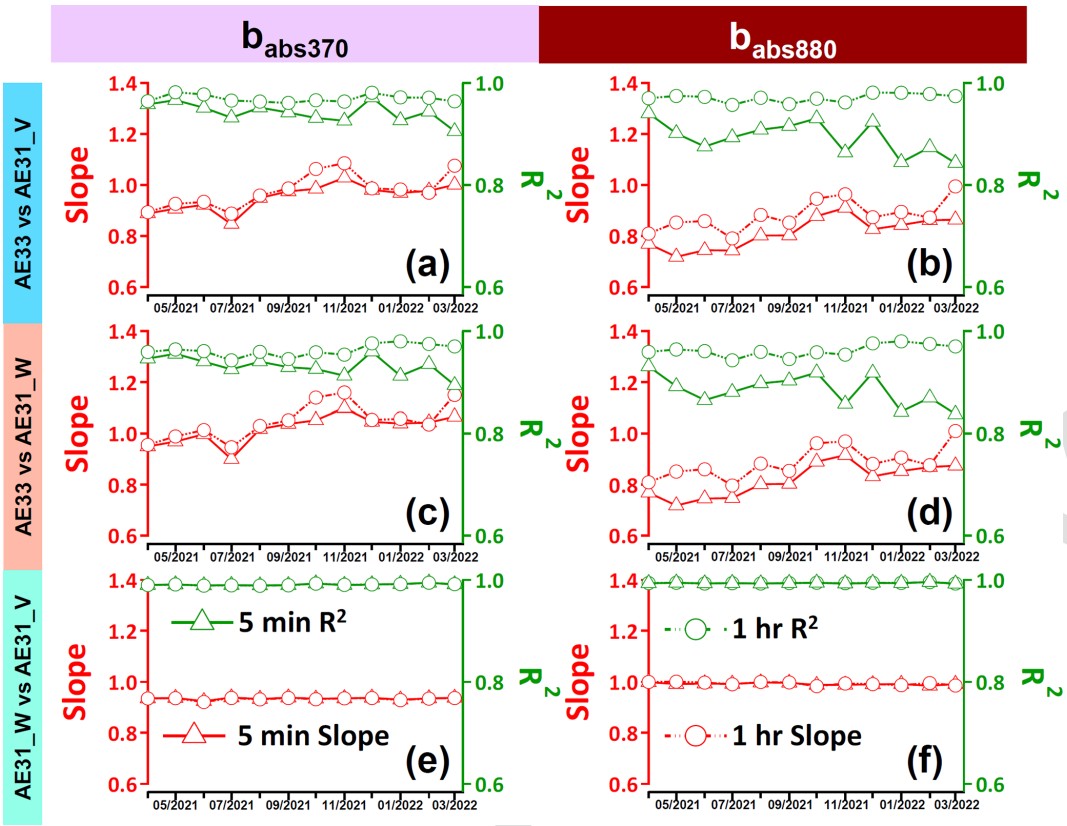

**Figure 4.** Monthly variation in slope and $R^2$ of AE33 and AE31 comparisons at 5 min and 1 h time resolutions.

**Table 3.** The absorption Ångström exponent (AAE) of AE31 and AE33 Aethalometers at 5 min and 1 h time bases. In the information given in the AAE subscripts, "/" denotes the AAE value calculated by the light absorption coefficients of two wavelengths and "−" denotes the AAE value obtained from the curve fitting of the power function using all wavelengths within the said range.

| AAE | Time base | AE31_V | AE31_W | AE33 |
|---|---|---|---|---|
| $AAE_{470/660}$ | 1 h | $1.12 \pm 0.15$ | $1.08 \pm 0.17$ | $1.33 \pm 0.12$ |
|  | 5 min | $1.12 \pm 0.34$ | $1.09 \pm 0.35$ | $1.34 \pm 0.14$ |
| $AAE_{370/880}$ | 1 h | $1.11 \pm 0.14$ | $1.05 \pm 0.17$ | $1.20 \pm 0.13$ |
|  | 5 min | $1.12 \pm 0.24$ | $1.05 \pm 0.26$ | $1.20 \pm 0.14$ |
| $AAE_{880/950}$ | 1 h | $1.75 \pm 0.47$ | $1.94 \pm 0.46$ | $0.67 \pm 0.12$ |
|  | 5 min | $1.81 \pm 1.02$ | $1.98 \pm 1.03$ | $0.68 \pm 0.22$ |
| $AAE_{370/950}$ | 1 h | $1.16 \pm 0.14$ | $1.12 \pm 0.17$ | $1.15 \pm 0.12$ |
|  | 5 min | $1.17 \pm 0.28$ | $1.12 \pm 0.29$ | $1.15 \pm 0.13$ |
| $AAE_{370-950}$ | 1 h | $1.11 \pm 0.12$ | $1.07 \pm 0.15$ | $1.19 \pm 0.12$ |
|  | 5 min | $1.14 \pm 0.27$ | $1.10 \pm 0.28$ | $1.19 \pm 0.13$ |

The inter-instrument $R^2$ values of AAEs were lower than those of $b_{abs}$ and eBC, as shown in Fig. 5. In general, the AAEs of AE31 data by Virkkula correction correlate better with AE33 data than with Weingartner correction. For example, for $AAE_{370/880}$, the $R^2$ of AE33 vs. AE31_V (0.56, Fig. 5d) was higher than the $R^2$ of AE33 vs. AE31_W (0.40, Fig. 5e). The best inter-instrument AAE agreement was observed in $AAE_{370/880}$, $AAE_{370/950}$ and $AAE_{370-950}$ for the AE33 vs. AE31_V comparison, with an $R^2$ of 0.56, 0.52 and

0.63, respectively. The AAE values obtained from 5 min and 1 h data were almost identical (Figs. S7 and S8 in the Supplement and Table 3), implying that AAE determination is not sensitive to the time resolution of $b_{abs}$ data.

A previous study by Zhang et al. (2019) suggested that $AAE_{880/950}$ can be used to represent the AAE of BC from fossil-fuel combustion ($AAE_{BC}$) for the AE33 data. The feasibility of this approach for AE31 data had not been examined. We found a significant disagreement of $AAE_{880/950}$ be-

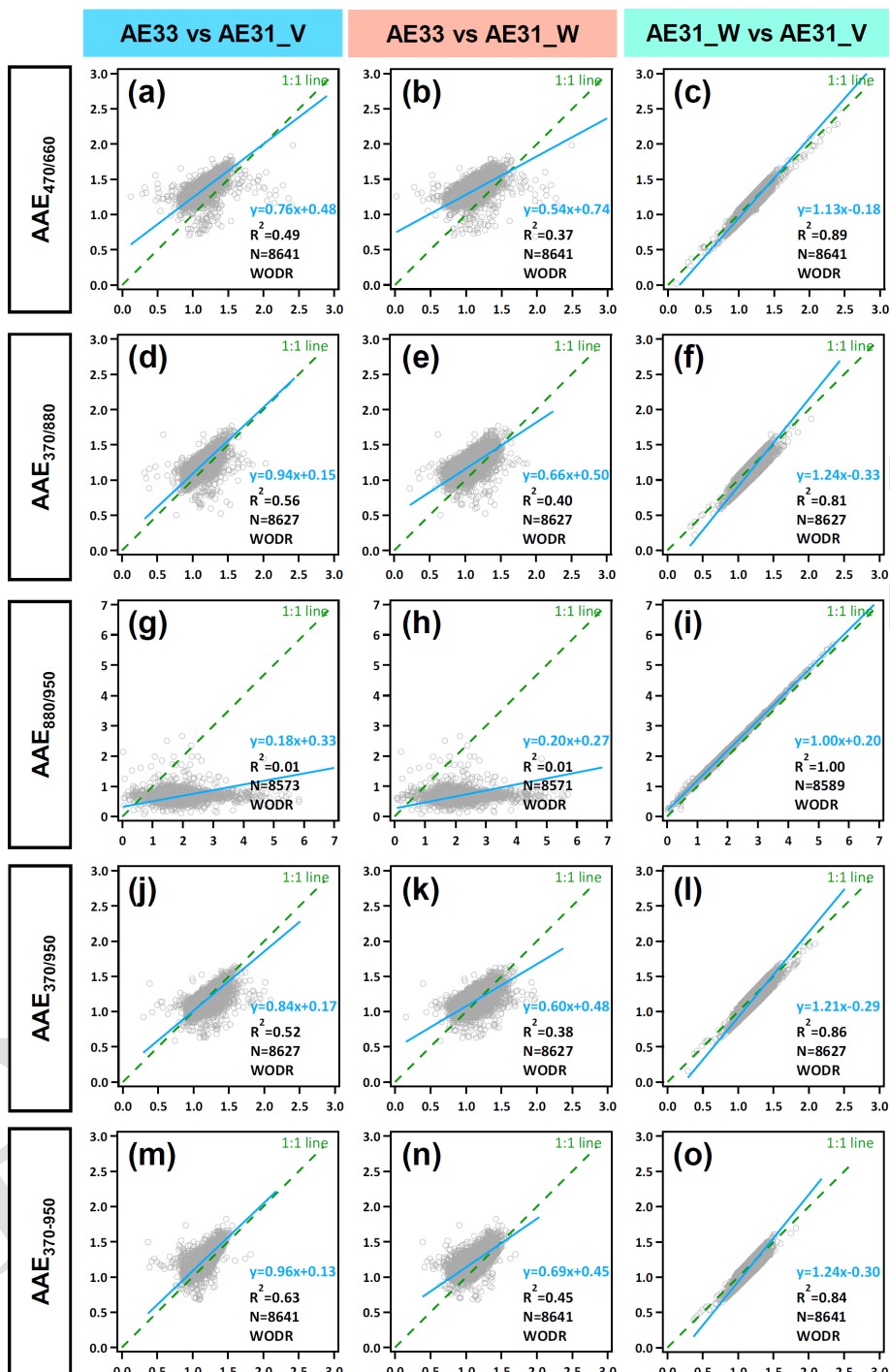

**Figure 5.** Inter-instrument comparison of different AAE values ($AAE_{470/660}$, $AAE_{370/880}$, $AAE_{880/950}$, $AAE_{370/950}$ and $AAE_{370-950}$) calculated using 1 h data from AE31_V, AE31_W and AE33.

tween AE33 and AE31 data, as indicated by the poor $R^2$ values (0.01, Fig. 5g and h) and diametrically different annual averages (1.75–1.98 for AE31 and 0.67–0.68 for AE33, Table 3). The discrepancy in $AAE_{880/950}$ between AE33 and AE31 may be associated with the difference in instrument design and filter material. Currently, PAS instruments that

measure at wavelengths of 880 and 950 nm do not exist. So there is no relevant literature to directly prove the inaccuracy of $AAE_{880/950}$ by AE31. A number of indirect clues reveal that $AAE_{880/950}$ by AE31 is less credible than $AAE_{880/950}$ by AE33. It is widely recognized that the AAE of BC from fossil-fuel combustion is close to unity (Bond

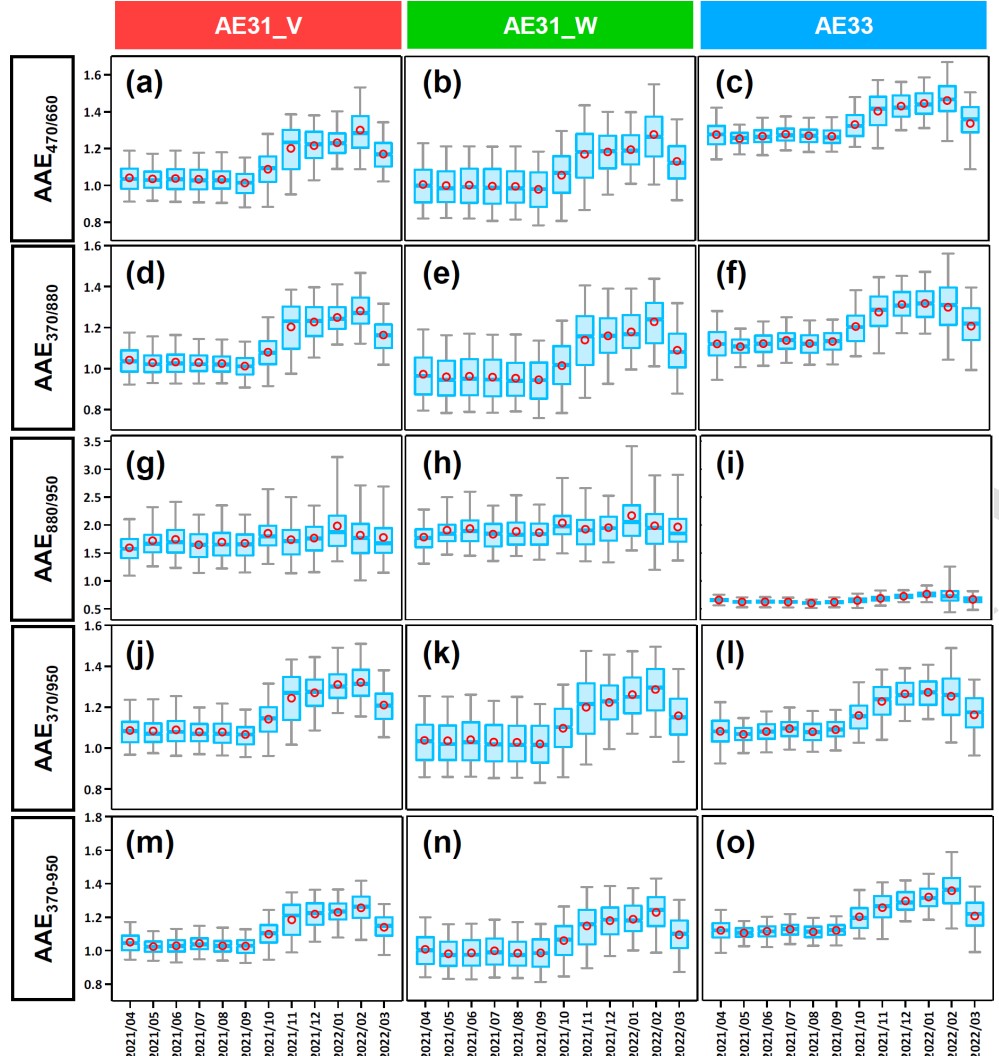

**Figure 6.** Monthly variations in different AAE values (AAE$_{470/660}$, AAE$_{370/880}$, AAE$_{880/950}$, AAE$_{370/950}$ and AAE$_{370-950}$) calculated using 1 h data from AE31_V, AE31_W and AE33.

and Bergstrom, 2006). The AAE$_{880/950}$ by AE31 ($\sim 2$, Table 3) is simply too high to represent AAE$_{BC}$. In contrast, AAE$_{880/950}$ by AE33 ($\sim 0.7$, Table 3) is much closer to the theoretical AAE$_{BC}$ (0.7–1) (Li et al., 2019; Liu et al., 2018) and also in agreement with filed measurements of AAE$_{BC}$ in Shenzhen (0.82–0.86) (Yuan et al., 2016), Beijing ($0.56 \pm 0.04$) (Wu et al., 2021), London (0.96) (Fuller et al., 2014), Wuhan (1.09) (Zheng et al., 2021) and Xi'an (1.19) (Wang et al., 2021a). Another piece of evidence is the distinct monthly variations in AAE as shown in Fig. 6 and Table S9 in the Supplement. During the wet season (April to September), the prevailing wind of the PRD is dominated by the oceanic air masses from the south; thus the AAE values are close to 1 (Fig. 6) as the ambient samples were dominated by local emissions. During the dry season (October to March), north wind prevails and the PRD was influenced by the long-range-transport air masses. As a result,

elevated AAE values were observed during the dry season due to the influence of biomass burning and coal combustion from eastern and northern China. Among all the AAE values (AAE$_{470/660}$, AAE$_{370/880}$, AAE$_{880/950}$, AAE$_{370/950}$ and AAE$_{370-950}$), AAE$_{880/950}$ is the only one that lacks seasonality (Fig. 6), confirming that AAE$_{880/950}$ can represent the AAE of BC from fossil-fuel combustion (AAE$_{BC}$) and is not affected by biomass burning during the dry season. These results suggest that AAE$_{BC}$ determination by AAE$_{880/950}$ is suitable for AE33 data but not suitable for AE31 data.

### 3.4 Comparison of secondary brown carbon light absorption estimation

Secondary brown carbon light absorption ($b_{abs\_BrCsec}$) estimation by the MRS approach has been widely adopted in recent studies (Wang et al., 2019a; Liakakou et al.,

**Table 4.** Summary of secondary brown carbon light absorption reported in the literature.

| Location | Model | Sampling period | Secondary brown carbon light absorption ($b_{abs\_BrCsec}$) (Mm$^{-1}$) | | | | | Reference |
|---|---|---|---|---|---|---|---|---|
| | | | 370 nm | 470 nm | 520 nm | 590 nm | 660 nm | |
| Guangzhou, China, urban site | AE33 | Apr 2021–Mar 2022 | $1.99 \pm 1.97$ | $0.91 \pm 0.84$ | $0.67 \pm 0.59$ | $0.45 \pm 0.41$ | $0.27 \pm 0.24$ | This study |
| | | Dry season | $2.34 \pm 2.08$ | $1.12 \pm 0.86$ | $1.05 \pm 0.69$ | $0.56 \pm 0.43$ | $0.34 \pm 0.26$ | |
| | | Wet season | $0.99 \pm 1.15$ | $0.38 \pm 0.45$ | $0.40 \pm 0.29$ | $0.23 \pm 0.23$ | $0.13 \pm 0.12$ | |
| | AE31_V | Apr 2021–Mar 2022 | $2.16 \pm 2.02$ | $1.10 \pm 0.97$ | $0.72 \pm 0.60$ | $0.45 \pm 0.37$ | $0.37 \pm 0.34$ | |
| | | Dry season | $2.68 \pm 2.15$ | $1.38 \pm 1.03$ | $0.93 \pm 0.64$ | $0.57 \pm 0.39$ | $0.46 \pm 0.37$ | |
| | | Wet season | $1.12 \pm 1.15$ | $0.55 \pm 0.52$ | $0.39 \pm 0.34$ | $0.24 \pm 0.21$ | $0.22 \pm 0.20$ | |
| Athens, Greece, urban site | AE33 | May 2015–Apr 2019 | $2.77 \pm 17.44$ | $0.69 \pm 4.94$ | $0.61 \pm 3.63$ | $0.16 \pm 1.25$ | $0.47 \pm 1.73$ | Liakakou et al. (2020) |
| Xianghe, China, rural site | AE33 | Dec 2017–Jan 2018 | 11.8 | 8.8 | 6.2 | 4.3 | 3.3 | Wang et al. (2019b) |
| Wuhan, China, urban site | AE31 | Jan 2020 | $4.9 \pm 4.6$ | – | – | – | – | Wang et al. (2021b) |
| Xi'an, China, urban site | AE33 | Nov 2015–Feb 2016 | 34.9 | 11.4 | 5.6 | 3.5 | 2.3 | Zhu et al. (2021b) |
| Xi'an, China, urban site | AE31 | 16 Dec 2016–15 Jan 2017 | 25.8 | 4.0 | 3.7 | 2.4 | 1.4 | Zhang et al. (2020) |
| Hong Kong SAR, China, urban site | | 16 Dec 2016–15 Jan 2017 | 4.8 | 3.4 | 2.4 | 1.7 | 1.2 | |
| Qinghai Lake, China, rural site | AE33 | Nov 2019–Feb 2020 | 7.9 | 2.2 | 1.0 | 0.6 | 0.3 | Zhu et al. (2021a) |
| Shaanxi, China, Mount Hua | AE33 | Aug 2018 | $4.4 \pm 6.1$ | – | – | – | – | Gao et al. (2022) |
| Guanzhong Plain, China, rural site | AE31 | Dec 2015–Jan 2016 | 3.1 | 1.5 | 0.3 | 0.1 | 0.4 | Qu et al. (2023) |
| Brisbane, Australia urban site | AE31 | Jul–Sep 2022 | 1.4 | 0.6 | 0.5 | 0.2 | 0.3 | Wu et al. (2023) |
| Tibetan Plateau, China, rural site | AE33 | Mar–May 2018 | 6.9 | 5.7 | 4.1 | 3.6 | 2.1 | Wang et al. (2019a) |
| Nanjing, China, urban site | AE33 | Jan–Mar 2020 | $3.7 \pm 4$ | $2.0 \pm 2$ | $1.7 \pm 1$ | $1.0 \pm 0.8$ | $0.4 \pm 0.5$ | Lin et al. (2021) |

2020; Zhu et al., 2021b; Wu et al., 2023). To date, the difference in $b_{abs\_BrCsec}$ determination between AE31 and AE33 has not been reported. With year-long collocated AE33 and AE31 data, this study aims to investigate the inter-instrument agreement on $b_{abs\_BrCsec}$. The annual average values of $b_{abs\_BrCsec}$ (1 h data) obtained in this study were $2.16 \pm 2.02$ Mm$^{-1}$ (AE31_V), $2.61 \pm 2.35$ Mm$^{-1}$ (AE31_W) and $1.99 \pm 1.97$ Mm$^{-1}$ (AE33), as shown in Table S10 in the Supplement. For AE31 results, $b_{abs\_BrCsec}$ by Weingartner correction is higher than that by Virkkula correction. This result suggests that secondary brown carbon light absorption estimation is sensitive to the data correction algorithm. Since the $b_{abs\_BrCsec}$ of AE31_V agrees better

with AE33 (Fig. S9a in the Supplement, $R^2 = 0.72$) than AE31_W (Fig. S9b, $R^2 = 0.44$), further comparisons are focused on AE31_V vs. AE33. As shown in Fig. 7a, the linear regression of AE33 vs. AE31 yields a slope of 0.78 and close-to-zero intercept ($-0.04$). The annual difference in the arithmetic mean in $b_{abs\_BrCsec}$ is 13 %. Most of the monthly difference in the arithmetic mean in $b_{abs\_BrCsec}$ is within 20 % (Fig. 7b), except for in May 2021 (39 %). These results suggest that despite the monthly difference in the arithmetic mean in $b_{abs\_BrCsec}$ being typically $\sim 20$ %, the $b_{abs\_BrCsec}$ between AE31 and AE33 is highly correlated and comparable.

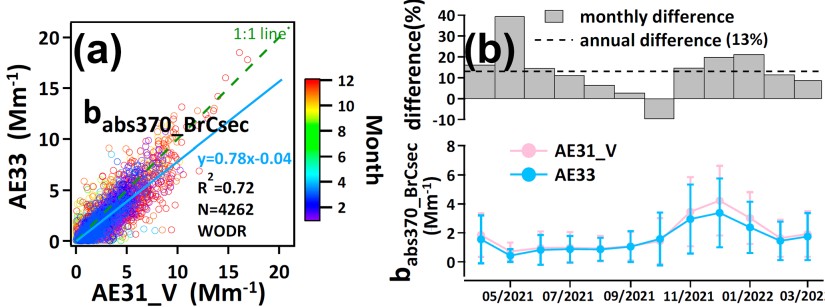

**Figure 7.** Comparison of secondary brown carbon light absorption at 370 nm ($b_{\text{abs370\_BrCsec}}$) estimated by AE33 and AE31_V (AE31 data correction using Virkkula algorithm). **(a)** Scatterplot of hourly $b_{\text{abs370\_BrCsec}}$ comparison between AE33 and AE31_V. The color coding represents months. **(b)** Monthly comparison of $b_{\text{abs370\_BrCsec}}$ between AE33 and AE31_V.

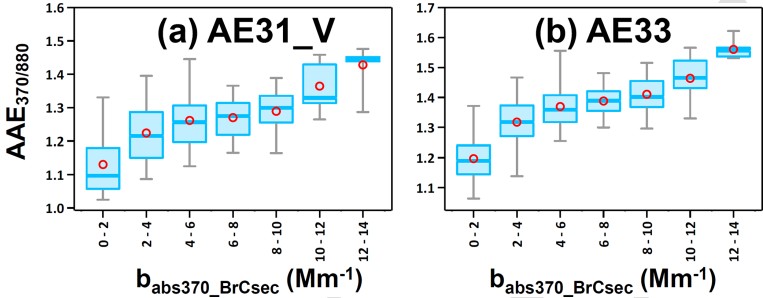

**Figure 8.** Dependence of $\text{AAE}_{370/880}$ on $b_{\text{abs370\_BrCsec}}$. $\text{AAE}_{370/880}$ is the absorption Ångström exponent calculated from the light absorption at 370 and 880 nm, while $b_{\text{abs370\_BrCsec}}$ is the secondary brown carbon light absorption at 370 nm. These panels were visualized using the Igor-based toolkit "Histbox" (Wu et al., 2018).

As summarized in Table 4, both AE31 and AE33 have frequently been used in the literature, and the $b_{\text{abs\_BrCsec}}$ found in this study is similar to those values found in Athens, Greece ($2.77 \pm 17.44\,\text{Mm}^{-1}$) (Liakakou et al., 2020), but lower than those found in Wuhan ($4.9\,\text{Mm}^{-1}$) (Wang et al., 2021b) and Xi'an ($34.9\,\text{Mm}^{-1}$) (Zhu et al., 2021b), China. Similarly to the monthly variations in AAE, $b_{\text{abs\_BrCsec}}$ exhibits distinct seasonality which is high in the dry season and low in the wet season. As shown in Fig. 8, a positive dependency of $\text{AAE}_{370/880}$ on $b_{\text{abs\_BrCsec}}$ was observed, suggesting that the AAE of aerosols was strongly affected by the abundance of secondary brown carbon. The diurnal pattern of $b_{\text{abs\_BrCsec}}$ obtained in this study, which is low in the daytime and high at nighttime (Fig. S10 in the Supplement), is similar to patterns of previous studies (Wang et al., 2019a; Q. Zhang et al., 2021). It is widely accepted that the daytime decrease in $b_{\text{abs\_BrCsec}}$ is largely associated with the photobleaching of brown carbon (Zhong and Jang, 2011; Li et al., 2023).

## 4  Conclusions and recommendations

A year-long collocated measurement comparison of a single-spot Aethalometer (AE31) and a dual-spot Aethalometer (AE33) was conducted in urban Guangzhou between 1 April 2021 and 31 March 2022. To minimize the interference of the filter loading effect, two data correction algorithms (Virkkula and Weingartner) were included in the comparison for AE31 data, while the instrument-embedded dual-spot correction was adopted for AE33 data. The main findings and recommendations of this study are summarized as follows.

The eBC detection limits of AE33 were largely improved compared to AE31 (e.g., $\text{LOD}_{\text{eBC\_AE33}} = 14.97\,\text{ng}\,\text{m}^{-3}$ vs. $\text{LOD}_{\text{eBC\_AE31}} = 52.2\,\text{ng}\,\text{m}^{-3}$ at 1 h, 880 nm). The improvement was more pronounced at high time resolutions (e.g., $\text{LOD}_{\text{eBC\_AE33}} = 96.57\,\text{ng}\,\text{m}^{-3}$ vs. $\text{LOD}_{\text{eBC\_AE31}} = 730.68\,\text{ng}\,\text{m}^{-3}$ at 2 min, 880 nm).

The eBC mass concentrations of the AE33 and AE31 were well correlated, with an $R^2$ of 0.97 and a slope of 1.20. The $\sim 20\,\%$ bias in the slope found in this study suggests that the AE33 / AE31 slope could be site-specific. To maintain the consistency of the historical data of AE31, the eBC mass concentration of AE33 Aethalometer was further adjusted by a second correction factor ($C_{\text{eBC}}$), which is the slope obtained in the comparison. The annual mean eBC values obtained from 1 h AE33 data were $2.35 \pm 1.37$ and $1.96 \pm 1.14\,\mu\text{g}\,\text{m}^{-3}$, respectively, before and after eBC post-adjustment. The later value agrees well with the AE31 annual average ($1.95 \pm 1.12\,\mu\text{g}\,\text{m}^{-3}$), and the AE33 vs. AE31 slope achieves 1.00 after eBC post-adjustment of AE33.

By adopting the localized multi-scattering correction factor ($C_{AE31} = 3.48$, $C_{AE33} = C_0 \times H = 1.39 \times 2.1 = 2.919$) obtained from previous studies, the $b_{abs}$ of AE33 agrees well with AE31, as evidenced by the close-to-unity regression slope and high $R^2$ of the 1 h data. The $b_{abs}$ agreement slightly varies by wavelength (slope: 0.87–1.04; $R^2$: 0.95–0.97) and by month, but such $b_{abs}$ agreement variations are not sensitive to the correction schemes (Virkkula or Weingartner) for AE31.

A variety of AAE values ($AAE_{470/660}$, $AAE_{370/880}$, $AAE_{880/950}$, $AAE_{370/950}$ and $AAE_{370-950}$) calculated using data from AE31_V, AE31_W and AE33 were compared. The AAE values are moderately correlated between AE33 and AE31 ($R^2$: 0.37–0.63), except for $AAE_{880/950}$. It is suggested that $AAE_{880/950}$ can be used for $AAE_{BC}$ estimation (Zhang et al., 2019). The $AAE_{880/950}$ of AE31 found in this study is too high and is not correlated with that of AE33 ($R^2 = 0.01$). These results suggest that $AAE_{BC}$ determination by $AAE_{880/950}$ is suitable for AE33 data but not suitable for AE31 data.

Secondary brown carbon light absorption ($b_{abs\_BrCsec}$) estimation by the MRS approach is sensitive to the data correction algorithm for AE31 results, and $b_{abs\_BrCsec}$ by the Weingartner correction is higher than that by the Virkkula correction. It is found that $b_{abs\_BrCsec}$ of AE31_V agrees better with AE33 than AE31_W. The annual difference in the arithmetic mean in $b_{abs\_BrCsec}$ between AE33 and AE31_V is 13 %, with an $R^2$ of 0.72. Despite the monthly difference in arithmetic means in $b_{abs\_BrCsec}$ being typically $\sim 20$ %, the $b_{abs\_BrCsec}$ between AE31 and AE33 is highly correlated and comparable, but such inter-instrument difference is not negligible and should be taken into account for secondary brown carbon estimation.

To ensure data continuity in long-term Aethalometer measurements when transitioning from the older (AE31) to the newer (AE33) model in permanent global-climate and air-quality stations, site-specific $C_{ref}$ and eBC correction factors are needed.

*Code availability.* The "Scatter Plot" toolkit (Wu and Yu, 2018) can be found at https://doi.org/10.5281/zenodo.832416 (Wu, 2020a). The "Histbox" toolkit (Wu et al., 2018) can be found at https://doi.org/10.5281/zenodo.832405 (Wu, 2020b). The "Aethalometer data processor" toolkit (Wu et al., 2018) can be found at https://doi.org/10.5281/zenodo.832403 (Wu, 2020c).

*Data availability.* The data used in this study are available upon request from the corresponding author (wucheng.vip@foxmail.com).

*Supplement.* The supplement related to this article is available online at: https://doi.org/10.5194/amt-17-1-2024-supplement.

*Author contributions.* LW: formal analysis, investigation, data curation, visualization, writing (original draft). CW: conceptualization, methodology, formal analysis, investigation, data curation, visualization, writing (original draft), writing (review and editing), supervision, project administration, funding acquisition. TD: resources. DW: resources, funding acquisition. ML: resources. YJL: writing (review and editing), funding acquisition. ZZ: resources, funding acquisition.

*Competing interests.* The contact author has declared that none of the authors has any competing interests.

ther geographical representation in this paper. While Copernicus Publications makes every effort to include appropriate place names, the final responsibility lies with the authors.

*Acknowledgements.* The authors would like to express their gratitude to Rigler Martin from Magee Scientific Co./Aerosol d.o.o. for providing valuable feedback during the preparation of the manuscript.

*Financial support.* This research has been supported by the National Natural Science Foundation of China (grant no. 42377089), the Fundo para o Desenvolvimento das Ciências e da Tecnologia (file no. 0023/2021/A1), the University of Macau (grant no. MYRG2022-00027-FST), and the Special Fund Project for Science and Technology Innovation Strategy of Guangdong Province (grant no. 2019B121205004).

*Review statement.* This paper was edited by Yuanjian Yang and reviewed by two anonymous referees.

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

**Remarks from the typesetter**