# Peer review of "Field comparison of dual- and single-spot aethalometers: Equivalent black carbon, light absorption, Ångström exponent and secondary brown carbon estimations"

_Atmospheric Measurement Techniques, 2023_

## Author Comment (AC1)

**Point-by-point response to reviewers' comments (amt-2023-245)**

**Dear Editor and Referees:**

On behalf of the co-authors, we thank the editor and the reviewers for their time and constructive comments on manuscript improvement. Here we provide a point-by-point response to those comments and make changes accordingly. In this document, the author's response is in purple, and the added/revised text is in blue. In the revised manuscript, the changes made are also in blue.

**Reviewer 1:**

Reviewer #1: This study compared the different parameters retrieved by two model Aethalometers (AE31 and AE33) based on one long-term observation. These parameters are important to study the environmental and climate effects of absorbing aerosols. I recommend the publication of this paper after the following comments are resolved.

Response: Thank you very much for your valuable suggestions. We have revised the manuscript accordingly.

Major comment:

This study only simply compares the measurement values of two model Aethalometers. The retrieved parameters are highly dependent on the transformation parameters. These transformation parameters are artificial, which is important for the final comparison results. Therefore, the sensitivity test of these transformation parameters is needed. In addition, this study should give a conclusion about which model Aethalometer measures more accurately. Therefore, comparing two Aethalometers with other instruments (such as PAS) is needed to solve this problem.

**Response to R1Q1:** Thank you for your constructive comments and helpful suggestions. We fully agreed with the reviewer that a parallel comparison of the Aethalometer with an in-situ reference instrument (e.g., PAS) is crucial for accurate $b_{abs}$ determination by the Aethalometer for a specific location. Since PAS was not available at our sampling site during the study period, we used C values from previous comparison studies conducted by our group in this region (Guangzhou). The C = 3.48 for AE31 was based on our previous intercomparison study between aethalometer and PAS in Guangzhou (Wu et al., 2013), as mentioned in the main text and Table S2. The C = 2.919 for AE33 was based on our previous intercomparison study between aethalometer and CRD-Neph in Guangzhou (Qin et al., 2018). Although the two C values were not determined during the same period, the corresponding $b_{abs}$ agree well in this study, as evidenced by the close-to-unity slopes (0.87~1.04) shown in Figure 4 (Figure 3 in the revised manuscript). A sensitivity test on C values was added. According to a recent comparison study between PAS and Aethalometer (Zhao et al., 2020), the C value deviation was found to be ± 0.4 in North China Plain. Thus, a deviation of ± 0.4 with an interval of 0.1 was used for the sensitivity test. The results are shown below (also in Table S8). With a deviation of ± 0.4 for the C values, the corresponding AE33/AE31 slopes of $b_{abs}$ range from 0.81 to 1.11, which provides a rough estimation of $b_{abs}$ uncertainty due to C values. It is also worth noting that the uneven agreement

of $b_{abs}$ between different wavelengths. This issue is related to the use of a single C value across all seven wavelengths, which is due to the absence of multi-wavelength PAS for covering 370 to 950 nm. This issue cannot be fully resolved before the emergence of PAS for covering 370 to 950 nm.

The following contents are added.

Lines 381 – 392

To explore the $b_{abs}$ uncertainty due to C values, a sensitivity test on different C values was performed. According to a recent comparison study between PAS and Aethalometer (Zhao et al., 2020), the C value deviation was found to be ± 0.4 in North China Plain. Thus, a deviation of ± 0.4 with an interval of 0.1 was used for the sensitivity test. The results are shown in Table S8. With a deviation of ± 0.4 for the C values of AE33, the corresponding AE33/AE31 slopes of $b_{abs}$ range from 0.81 to 1.11, which provides a rough estimation of $b_{abs}$ uncertainty due to C values. It is also worth noting the uneven agreement of $b_{abs}$ between different wavelengths. This issue is related to the use of a single C value across all seven wavelengths, which is due to the absence of multi-wavelength PAS. By far, the wavelength coverage of PAS instruments remains limited, e.g. two wavelengths (405 and 880 nm)(Lewis et al., 2008), and four wavelengths (405 to 660 nm)(Schnaiter et al., 2023). This issue cannot be fully resolved before the emergence of PAS that can fully cover the wavelengths of Aethalometer from 370 to 950 nm.

**Table S8.** Sensitivity test of C value on the comparison of AE33 vs. AE31. The value of $C_{AE33}$ was varied from 2.419 to 3.319 with an interval of 0.1 while $C_{AE31}$ was set to a constant (3.48) during the sensitivity test.

| $C_{AE33}$ | AE33/AE31 slope | | | | | | | |
| --- | --- | --- | --- | --- | --- | --- | --- | --- |
| | 370 nm | 470 nm | 520 nm | 590 nm | 660 nm | 880 nm | 950 nm | Average |
| 2.419 | 1.17 | 1.17 | 1.05 | 1.13 | 1.08 | 1.04 | 1.13 | 1.11 |
| 2.519 | 1.12 | 1.13 | 1.01 | 1.09 | 1.03 | 1.00 | 1.08 | 1.07 |
| 2.619 | 1.08 | 1.08 | 0.97 | 1.04 | 0.99 | 0.96 | 1.04 | 1.02 |
| 2.719 | 1.04 | 1.04 | 0.93 | 1.01 | 0.96 | 0.93 | 1.00 | 0.99 |
| 2.819 | 1.00 | 1.01 | 0.90 | 0.97 | 0.92 | 0.90 | 0.97 | 0.95 |
| 2.919 | 0.97 | 0.97 | 0.87 | 0.94 | 0.89 | 0.87 | 0.93 | 0.92 |
| 3.019 | 0.93 | 0.94 | 0.84 | 0.91 | 0.86 | 0.84 | 0.90 | 0.89 |
| 3.119 | 0.90 | 0.91 | 0.81 | 0.88 | 0.84 | 0.81 | 0.87 | 0.86 |
| 3.219 | 0.88 | 0.88 | 0.79 | 0.85 | 0.81 | 0.79 | 0.85 | 0.84 |
| 3.319 | 0.85 | 0.85 | 0.76 | 0.82 | 0.78 | 0.76 | 0.82 | 0.81 |

Detailed comments:

1. Line 75. What is the special significance of one year of observation? The authors didn't talk about intercomparing seasonality in the following discussion.

**Response to R1Q2**: We chose to conduct a one-year study is to account for all the possible seasonality of $b_{abs}$ and BC mixing state, which is absent in previous studies as summarized in Table 1. The seasonality of the comparison was discussed in the original manuscript. As shown in Fig. 5 and Table S7, AE33/AE31 slopes of $b_{abs}$ varied by month and the corresponding discussions can be found in section 3.2. The seasonality of AAE agreement was illustrated in Figure 7 and discussed in section 3.3. The monthly variations of the secondary brown carbon light absorption is compared in Figure 8, and Table S9 and discussed in section 3.4.

2. Line 100. The flow rate of AE33 is generally set as 5 Lpm. Whether this flow difference will cause a difference in the comparison result?

**Response to R1Q3**: The default flow rate of AE33 and AE31 is 5 Lpm but needs to be adjusted according to the sampling environment. As shown in Table 1, the flow rate of previous AE33/AE31 comparison studies ranged from 3 to 5.8 Lpm. A lower flow rate can increase the LOD and that could be an issue for the background sites (e.g., polar regions). Since the eBC concentration in the urban environment is much higher than the LOD of Aethalometer, the impact is expected to be neglectable. To clarify why the flow rate was set lower than 5 Lpm, the following contents were added.

Lines 101 to 109

Both AE31 and AE33 were connected to one $PM_{2.5}$ inlet (Figure S1). An inline Nafion dryer (MD-700, Perma Pure, NJ, USA) was used to minimize the impact of relative humidity. Due to the drying capacity of the Nafion dryer, the flow rate of the two Aethalometers was set to be lower than the default value (5 Lpm). A lower flow rate can increase the LOD and that could be an issue for the background sites (e.g., polar regions). Since the eBC concentration in the urban environment is much higher than the LOD of the Aethalometer, the impact is expected to be neglectable. The single-spot AE31 was operated at a flow rate of 2.4 Lpm using the quartz fiber filter tape (Pallflex, type Q250F). The dual-spot AE33 measurement was conducted at a flow rate of 3 Lpm using filter tape 8060.

[Figure]

**Figure S1**. Tubing connection diagram of AE33 and AE31.

**Point-by-point response to reviewers' comments (amt-2023-245)**

3. Line 105. What's the RH range of dried sample flow? More information about the sample needs to be provided.

**Response to R1Q4**: Thanks for the suggestions and the corresponding contents were added.

Lines 109 to 115

To evaluate the effectiveness of the Nafion dryer, the ambient RH and RH after the Nafion dryer were compared in Figure S2. The annual average RH was reduced from 60.50±13.24 to 45.59±1.12 %. More importantly, before drying, half of the data points had an RH higher than 60%, but after drying 95% of the data points had an RH lower than 60% (Figure S2a). Additionally, the diurnal fluctuations were effectively minimized after Nafion drying (Figure S2b). These results suggested that the RH of the sample air was well controlled before entering the two Aethalometers.

[Figure]

**Figure S2.** Ambient and dried sample RH (Nafion) characteristics during the one-year campaign. (a) Box plot of annual average RH. Black circles represent annual average concentrations. The line inside the box indicates the annual median concentration. The upper and lower boundaries of the box represent the 75th and the 25th percentile; the whisker above and below each box represents the 95th and 5th percentile. (b) Diurnal variations of ambient and Nafion RH.

4. Eq. 8 and Eq. 15. The values of Cref and C' have an expected influence on the comparison result. Different studies used different empirical values. The sensitivity analysis of these values on the comparison result is recommended.

**Response to R1Q4**: A sensitivity test on C values was added. More details can be found in the response to **R1Q1**.

5. Eq. 20. Compared to maintaining consistency, it's more important to acquire eBC accurately. This equation assumes the right measurement by AE31, and then correct the measurement by AE33. However, the measurement of AE33 may be more accurate because of the technology development. Therefore, it's more reasonable to correct AE31 data. More comparison of two Aethalometers with other instruments (such as PAS) is needed to solve this question.

**Response to R1Q5:** We fully agree with the reviewer that AE33 provides a higher

precision than AE31 as evidence by the smaller deviation shown in Figure 1. However, the accuracy of eBC determination depends on the conversion factor $\sigma_{ATN}$, which is obtained from the regression slope between $b_{ATN}$ and EC by the EGA (evolve gas analysis) method (Gundel et al., 1984). Considering the operational defined nature of eBC and the large number of historical eBC data obtained by legacy type Aethalometers, it would be more appropriate to align the eBC from the newer model to the eBC from legacy type Aethalometers. The following contents were added to improve the clarity.

Lines 136-143

Here $\sigma_{ATN}$ is the conversion factor between $b_{ATN}$ and eBC, which is obtained from the regression slope between $b_{ATN}$ and EC by the EGA (evolve gas analysis) method (Gundel et al., 1984). Developed by Lawrence Berkeley National Laboratory (LBNL), the EGA method (Ellis et al., 1984) was commonly used from the 1980s to 1990s (Ip et al., 1984; Turner and Hering, 1990; Young et al., 1994) and became less popular in recent years. Since the BC of Aethalometer was calibrated to the LBNL-EGA EC, differences in EC analysis protocols lead to a disagreement between eBC and other popular EC methods (e.g. NIOSH and IMPROVE) used nowadays. In general, it is recognized that eBC is usually higher than NIOSH (Jeong et al., 2004) but lower than IMPROVE (Watson and Chow, 2002).

Lines 322-327

Considering the operationally defined nature of eBC and the large amount of historical eBC data accumulated by legacy-type Aethalometers, it would be more appropriate to align the eBC from the newer model to the eBC from legacy-type Aethalometers to maintain the consistency of the historical data of AE31. For these reasons, the eBC mass concentration of AE33 Aethalometer was further adjusted by a second correction factor ($C_{eBC}$, 1.20 and 1.18 for 1 hr and 5 min data, respectively), which is the slope obtained in Figure 2:

**Reviewer 2:**

Reviewer #2: After AE31, Aethalometer model AE33 is also joining in monitoring black carbon (BC) mass concentration and light absorption coefficient ($b_{abs}$) around the world. This implies a need of field intercomparison of the two models. In this study, the authors carried out a year-long collocated measurement comparison of the two models in urban Guangzhou. Considering that field intercomparisons of the two models are limited and the difference in secondary brown carbon light absorption estimation between the two models is largely unknown, this study could add knowledge about the instruments and applications. Scholars around the world could learn much from the results of this study. I recommend publication of this work in AMT once my concerns, major or minor, have been properly addressed.

Response: We thank the reviewer for the positive comments and helpful suggestions.

**Specific comments:**

1. To compare the performance of the two aethalometer models, AE31 and AE33, a collocated observation campaign should begin with a carefully prepared state. It is not clear here whether the two instruments were status-checked. Another concern relates to whether the two instruments were calibrated with a reference instrument of absorption measurement (such as PAX, PASS, or MAAP), or at least flowrate-calibrated with a standard flow meter. Informing readers of the starting point helps understand the comparison results in this study.

**Response to R2Q1:** Thanks for the constructive comments and helpful suggestions. The setup of the collocated experiment was illustrated in newly added Figure S1 as shown below.

[Figure]

**Figure S1**. Tubing connection diagram of AE33 and AE31.

**Point-by-point response to reviewers' comments (amt-2023-245)**

QA/QC is crucial for obtaining reliable comparison results. The instrument maintenance procedures were added.

Lines 117 – 120

Routine maintenance procedures suggested by Cuesta-Mosquera et al. (2021) were implemented in this study. The optical chamber of the two Aethalometers was carefully cleaned before the collocated experiment and repeated every three months. Flow verification and calibrations of the two Aethalometers were conducted every three months using an external flow meter (Bios Defender 520H, Mesa Labs, CO, USA). Blank test and leak test were performed monthly for the two Aethalometers.

We fully agreed with the reviewer that a parallel comparison of the Aethalometer with an in-situ reference instrument (e.g., PAS) is crucial for accurate $b_{abs}$ determination by the Aethalometer for a specific location. Since PAS was not available at our sampling site during the study period, we used C values from previous comparison studies conducted by our group in this region (Guangzhou). The C = 3.48 for AE31 was based on our previous intercomparison study between aethalometer and PAS in Guangzhou (Wu et al., 2013), as mentioned in the main text and Table S2. The C = 2.919 for AE33 was based on our previous intercomparison study between aethalometer and CRD-Neph in Guangzhou (Qin et al., 2018). Although the two C values were not determined during the same period, the corresponding $b_{abs}$ agree well in this study, as evidenced by the close-to-unity slopes (0.87~1.04) shown in Figure 4 (Figure 3 in the revised manuscript). A sensitivity test on C values was added. According to a recent comparison study between PAS and Aethalometer (Zhao et al., 2020), the C value deviation was found to be ± 0.4 in North China Plain. Thus, a deviation of ± 0.4 with an interval of 0.1 was used for the sensitivity test. The results are shown below (also in Table S8). With a deviation of ± 0.4 for the C values, the corresponding AE33/AE31 slopes of $b_{abs}$ range from 0.81 to 1.11, which provides a rough estimation of $b_{abs}$ uncertainty due to C values. It is also worth noting that the uneven agreement of $b_{abs}$ between different wavelengths. This issue is related to the use of a single C value across all seven wavelengths, which is due to the absence of multi-wavelength PAS for covering 370 to 950 nm. This issue cannot be fully resolved before the emergence of PAS for covering 370 to 950 nm.

The following contents are added in the updated manuscript.

Lines 381 – 392

To explore the $b_{abs}$ uncertainty due to C values, a sensitivity test on different C values was performed. According to a recent comparison study between PAS and Aethalometer (Zhao et al., 2020), the C value deviation was found to be ± 0.4 in North China Plain. Thus, a deviation of ± 0.4 with an interval of 0.1 was used for the sensitivity test. The results are shown in Table S8. With a deviation of ± 0.4 for the C values of AE33, the corresponding AE33/AE31 slopes of $b_{abs}$ range from 0.81 to 1.11, which provides a rough estimation of $b_{abs}$ uncertainty due to C values. It is also worth noting the uneven agreement of $b_{abs}$ between different wavelengths. This issue is related to the use of a single C value across all seven wavelengths, which is due to the absence of multi-wavelength PAS. By far, the wavelength coverage of PAS instruments remains limited, e.g. two wavelengths (405 and 880 nm)(Lewis et al., 2008), and four wavelengths (405 to 660 nm)(Schnaiter et al., 2023). This issue cannot be fully resolved before the emergence of PAS that can fully cover the wavelengths of Aethalometer from 370 to 950 nm.

**Table S8.** Sensitivity test of C value on the comparison of AE33 vs. AE31. The value of $C_{AE33}$ was varied from 2.419 to 3.319 with an interval of 0.1 while $C_{AE31}$ was set to a constant (3.48) during the sensitivity test.

| $C_{AE33}$ | AE33/AE31 slope | | | | | | | |
|---|---|---|---|---|---|---|---|---|
| | 370 nm | 470 nm | 520 nm | 590 nm | 660 nm | 880 nm | 950 nm | Average |
| 2.419 | 1.17 | 1.17 | 1.05 | 1.13 | 1.08 | 1.04 | 1.13 | 1.11 |
| 2.519 | 1.12 | 1.13 | 1.01 | 1.09 | 1.03 | 1.00 | 1.08 | 1.07 |
| 2.619 | 1.08 | 1.08 | 0.97 | 1.04 | 0.99 | 0.96 | 1.04 | 1.02 |
| 2.719 | 1.04 | 1.04 | 0.93 | 1.01 | 0.96 | 0.93 | 1.00 | 0.99 |
| 2.819 | 1.00 | 1.01 | 0.90 | 0.97 | 0.92 | 0.90 | 0.97 | 0.95 |
| 2.919 | 0.97 | 0.97 | 0.87 | 0.94 | 0.89 | 0.87 | 0.93 | 0.92 |
| 3.019 | 0.93 | 0.94 | 0.84 | 0.91 | 0.86 | 0.84 | 0.90 | 0.89 |
| 3.119 | 0.90 | 0.91 | 0.81 | 0.88 | 0.84 | 0.81 | 0.87 | 0.86 |
| 3.219 | 0.88 | 0.88 | 0.79 | 0.85 | 0.81 | 0.79 | 0.85 | 0.84 |
| 3.319 | 0.85 | 0.85 | 0.76 | 0.82 | 0.78 | 0.76 | 0.82 | 0.81 |

2. Line 31 and some other lines: In the manuscript, site-specific correction was again and again proposed possibly because the authors tend to believe that some differences are caused by differed composition or property of monitored aerosols from different sites. This is only a possibility instead of a certainty considering many model-related factors like hardware, mechanism, or filter tape, may all make differences in results. For example, filter tapes from different manufacturers with different materials may give substantially differed information in scattering coefficient, AAE, and so on. From this perspective, authors of this manuscript are proposed to be cautious whenever attributing differences between AE33 and AE31 to site-specificness unless properly and fully evidenced. Particularly when you used two different models of aethalometers in the same site in this study. Check through the manuscript to revise improper descriptions.

**Response to R2Q2:** We fully agree with the reviewer that the differences between AE33 and AE31 were associated with factors like hardware, mechanism and filter type. Along with site-dependent aerosol type and mixing state, these factors lead to the site-dependent differences between AE33 and AE31. To clarify this point, the following contents are revised.

Lines 311 – 320

The eBC differences between AE33 and AE31 were associated with factors like hardware design and filter type. Along with site-dependent aerosol type and mixing state, these factors could lead to site-dependent eBC differences between AE33 and AE31. According to the technical notes of the manufacturer (Magee-Scientific, 2017), the slope of eBC by 8060/8020 filter varied by different locations, e.g., Beijing (0.82), Bangalore (0.87), Paris (0.93) and Berkeley (0.94). The filter used by AE31 (Quartz-fiber filter, Pallflex Q250F) is very different from the filter used by AE33 (8060) in terms of material and optical properties. Likewise, the site-dependent filter difference could contribute to the site-dependent AE33/AE31 difference. Along with the variations of AE33/AE31 difference reported in previous studies (Table 1), the ~20% bias in the slope found in this study suggests that the AE33/AE31 slope could be site-dependent.

3. Equation (15) and the paragraph below: The authors used C', $C_{AE33}$, and $C_{ref}$ to represent different meanings for scattering correction. The denominator of Equation (15) was used to correct for the bias generated by filter scattering. In this sense, C' is

anyway not appropriate because you have employed C' to represent "second correction factor". I suggest the authors keep unambiguity and consistency in the definition and application of the three forms of conversion factor and revise Equation (15).

**Response to R2Q3:** Thanks for the suggestions. The use of C' had been abandoned and the second correction factor was termed as "harmonization factor" according to a previous study (Savadkoohi et al., 2023).

Equation (15) was revised as follows:

$$b_{abs}\,(AE33) = \frac{eBC_{corrected} \cdot \sigma_{air}}{H} \tag{15}$$

4. Figure 3: Since you have customized the correction factor for filtering scattering ($C_{ref?}$) according to the differences in eBC between AE31V and AE33 so that equal eBC values from AE31V and AE33 could be yielded, it is not surprising that an extremely good correlation and slope (1:1) was observed after 2nd correction. For this reason, I don't think Figure 3 is of high value. You may move it to Supporting Information.

**Response to R2Q4:** Thanks for the suggestions. Figure 3 was moved to the supplement material.

5. Regarding toolkits: The authors mentioned a few toolkits for convenient data processing; quite helpful, I think. But for readers, little is known about the toolkits unless they go to read the articles the authors cited. Therefore I ask the authors to give a brief description of individual toolkits where the toolkit and associated articles are cited so that readers could know something about the principle and mechanism helpful for understanding output results.

**Response to R2Q5:** Thanks for the suggestions. We added the descriptions of toolkit in section 2.2.5 (also shown below).

Lines 246-267

Serval data analysis and visualization toolkits developed in our group were used in this study, including *Scatter Plot*, *Histbox*, and *Aethalometer Data Processor*.

*Scatter Plot*. Conventional ordinary least squares (OLS) assume that independent variables (X) are error-free. However, for inter-instrument comparison studies, X and Y (from two instruments) usually have comparable degrees of uncertainty. In this case, linear regression by OLS should be avoided as it leads to biased slope and intercept. To account for uncertainties in both X and Y, an error-in-variables linear regression technique, weighted orthogonal distance regression (WODR), was applied in this study, implemented by the Igor-based toolkit *Scatter Plot* (Wu and Yu, 2018). A free download of *Scatter Plot*

can be found at https://doi.org/10.5281/zenodo.832416.

*Histbox*. A handy tool enables batch plotting for histogram and box plots with specific optimization for atmospheric science (e.g. batch plotting by year/season/month, by hour, by day of week, by user-defined strings). The Igor-based *Histbox* toolkit (Wu et al., 2018) also provides data averaging and alignment functions which are common steps in atmospheric data processing (e.g. integrating data from various instruments with different time scales). Its comprehensive data sorting, grouping and screening features ensure efficient data visualization. A free download of *Histbox* can be found at https://doi.org/10.5281/zenodo.832405.

*Aethalometer data processor.* Data acquired from filter-based measurements such as legacy Aethalomenter (AE31/AE20) needs careful correction due to its inherent systemic error, i.e., filter matrix effect, scattering effect and loading effect. This toolkit (Wu et al., 2018) provides a user-friendly interface to implement Weingartner (2003) and Virkkula (2007) algorithms for Aethalometer data correction. QA/QC features are also provided, including statistics of sensor voltage. A free download of *Aethalometer data processor* can be found at https://doi.org/10.5281/zenodo.832403.

6. Lines 233-246: In this paragraph, the authors described LODs of eBC based on AE31 and AE33. Some of the results are easy to understand, e.g., longer time bases lead to lower LODs; but some results need a follow-up explanation. For example, the LODs are wavelength-dependent, why? Another example, AE33 has better LOD than AE31, why? Are there any new technologies having been incorporated in AE33 and thus contributed to the improvement in LOD? Do you know what technologies that work?

**Response to R2Q6:** The detection limits of Aethalometers are wavelength-dependent because the LED of each wavelength may have different characteristics in terms of light intensity stability, background noise, and detector response. The electronic can also affects the LOD of Aethalometer. A study on AE51 by Ning et al. (2013) showed that the LOD with 5V DC power supply was 5 times of LOD with battery power supply. The improved LOD of AE33 comparing to AE31 is the combination of advance in LED stability, flow control, optical chamber design and electronics (Drinovec et al., 2015).

7. Lines 284-285: "presumably because of the high loadings of light-absorbing aerosol particles at this urban site", why? Can you argue for it?

**Response to R2Q7:** The study by Asmi et al. (2021) reported a $b_{abs}$ level of approximately 0.1 Mm$^{-1}$, which is close to the limit of detection of Aethalometers and two orders of magnitude lower than the $b_{abs}$ level of the current study. It is not surprising, therefore, that the higher loading of light-absorbing aerosol particles in the current study results in a better agreement between AE33/AE31 for $b_{abs}$.

**Point-by-point response to reviewers' comments (amt-2023-245)**

8. Lines 354-355: "These results suggest that $AAE_{BC}$ determination by AAE880/950 is suitable for AE33 data but not suitable for AE31 data". Interesting, but why? Do you have some evidences (e.g., literatures) for this point?

**Response to R2Q7:** Currently, PAS with wavelengths of 880 nm and 950 nm do not exist. So there is no relevant literatures to directly prove the inaccuracy of $AAE_{880/950}$ by AE31. But we had discussed the indirect clues that $AAE_{880/950}$ by AE31 is less reasonable compared to $AAE_{880/950}$ by AE33. The $AAE_{880/950}$ by AE31 (~2, Table 3) is simply too high to represent $AAE_{BC}$. In contrast, $AAE_{880/950}$ by AE33 (~0.7, Table 3) is much closer to the theoretical $AAE_{BC}$ (0.7~1) (Li et al., 2019; Liu et al., 2018), and also in agreement with filed measurements of $AAE_{BC}$ in Shenzhen (0.82~0.86) (Yuan et al., 2016), Beijing (0.56±0.04) (Wu et al., 2021), Longdon (0.96) (Fuller et al., 2014), Wuhan (1.09) (Zheng et al., 2021) and Xi'an (1.19) (Wang et al., 2021).

To clarify this point, the following contents are revised.

Lines 416 – 425

Currently, PAS with wavelengths of 880 nm and 950 nm do not exist. So there is no relevant literature to directly prove the inaccuracy of $AAE_{880/950}$ by AE31. A number of indirect clues reveal that $AAE_{880/950}$ by AE31 is less credible than $AAE_{880/950}$ by AE33. It is widely recognized that the AAE of BC from fossil-fuel combustion is close to unity (Bond and Bergstrom, 2006). The $AAE_{880/950}$ by AE31 (~2, Table 3) is simply too high to represent $AAE_{BC}$. In contrast, $AAE_{880/950}$ by AE33 (~0.7, Table 3) is much closer to the theoretical $AAE_{BC}$ (0.7~1) (Li et al., 2019; Liu et al., 2018), and also in agreement with filed measurements of $AAE_{BC}$ in Shenzhen (0.82~0.86) (Yuan et al., 2016), Beijing (0.56±0.04) (Wu et al., 2021), Longdon (0.96) (Fuller et al., 2014), Wuhan (1.09) (Zheng et al., 2021) and Xi'an (1.19) (Wang et al., 2021).

**Technical corrections**

1. Line18-19: "However, field intercomparison of the two popular models, dual-spot (AE33) and single-spot (AE31) aethalometers, remain limited". Note the disagreement between subject and predicate in number. Please check through the manuscript against similar errors (e.g., in lines 280-281).

Response: Corrected.

2. Line 27: $BrC_{sec}$ is an abbreviated form. Give complete form first. Please also check through the text against similar errors.

Response: Secondary brown carbon ($BrC_{sec}$) is defined in line 20 in the original manuscript.

3. Line 88: Delete "was".

Response: Corrected.

4. Line 21, 66 collocated; Lines 82, 98, 383 colocated. Did you attempt to use "collocated" and "colocated" the same meaning?

Response: Thanks for the suggestion. "Collocated" is used throughout the manuscript

for consistency.

5. Line 100: Suggest inserting a hyphen between single and spot. L131: Add "where" before "$eBC_{corrected}$".

Response: Corrected.

6. Line 197: Change "Text" to "Test".

Response: "Text" is correct.

7. Lines 209-211: It seems better if you change "the $BrC_{sec}$ is generated during the secondary aging process" to "the $BrC_{sec}$ is secondarily generated during the aging process".

Response: Corrected as suggested.

8. Line 325: Attention, "fiiting" is ill-spelled.

Response: Typo corrected.

9. Line 363: "those" should be "that"; similar cases can be found in lines 410, 414.

Response: Corrected.

10. Line 378: obtain should be "obtained".

Response: Corrected.

11. Line 402: Revise "1.39· 2.1" to "1.39×2.1".

Response: Corrected.

12. Table 1: For AE31, there have been a column for "filter" scattering correction. why not add a column for loading correction?

Response: Added as suggested.

**Table 1.** Summary of existing AE31/AE33 intercomparison studies. W and V in the column "Loading correction" refer to Weingartner and Virkkula correction algorithms, respectively.

| Measurement site | Model | Time base | Flow rate (LPM) | Filter | Loading correction | Period (duration) | Slope (AE33 vs AE31) | Reference |
|---|---|---|---|---|---|---|---|---|
| Ahmedabad, India (urban) | AE31 | 5 min | 3 | Quartz fiber filter $C_{AE31}$= 2.14 | W | July 2014 – Dec 2014 | $eBC_{880}$ 5 min :1.06 | (Rajesh and Ramachandran, 2018) |
| | AE33 | 1 min | 3 | Teflon coated glass fiber $C_{AE33}$= 1.57 | Dual-spot | 6 months | 1 hr: 1.02 | |
| Milan, Italy (urban) | AE31 | 5 min | | Quartz fiber filter (Pallflex Q250F) | W | Jan. 18 – Feb. 15 (2018) | $eBC_{880}$ 5 min: 1.05 | (Ferrero et al., 2021) |
| | AE33 | 1 min | | Teflon-coated glass fiber (Pallflex T60A20) $C_{AE33}$= 1.57 | Dual-spot | 1 month | | |
| Granada, Spain (urban) | AE31 | 5 min | | | \ | June 2014 – July 2014 | $eBC_{880}$ 5 min: 1.11 | (Titos et al., 2015) |
| | AE33 | 1 min | | | Dual-spot | 2 months | | |

| | | | | | | | | |
|---|---|---|---|---|---|---|---|---|
| Pallas, Finland (background) | AE31 | 5 min | 4.5 | $C_{AE31}= 3.5$ | V | June. 19 – July. 17 (2019) | $b_{abs660}$ 1 hr: 0.47 | (Asmi et al., 2021) |
| | AE33 | 1 min | 5.8 | $C_{AE33}=1.39$ $C'=2.52$ | Dual-spot | 1 month | | |
| Guangzhou, China (urban) | AE31 | 5 min | 2.4 | Quartz-fiber filter (Pallflex Q250F) $C_{AE31}= 3.48$ | W&V | Apr 2021 – Mar 2022 | $eBC_{880}$ 5 min: 1.18 1 hr: 1.20 | This study |
| | AE33 | 1 min | 3 | M8060 $C_{AE33}=1.39$ $C'=2.1$ | Dual-spot | 12 months | $b_{abs880}$ 5 min: 0.85-0.86 1 hr: 0.87 | |

13. Figure S1: Attention to "change of of sampling spot" in the title.

Response: corrected.

**References**

[revised manuscript text omitted]